# Deep Learning and Machine Learning Modeling Identifies Thidiazuron as a Key Modulator of Somatic Embryogenesis and Shoot Organogenesis in *Ferula assa-foetida* L.

**DOI:** 10.3390/biology14121703

**Published:** 2025-11-29

**Authors:** Khushbu Kumari, Samaksh Mittal, Kritika Sharma, Sanatsujat Singh, Jyoti Upadhyay, Vishal Acharya, Virender Kadyan, Sudesh Kumar Yadav, Rohit Joshi

**Affiliations:** 1CSIR—Institute of Himalayan Bioresource Technology (CSIR-IHBT), Palampur 176061, Himachal Pradesh, India; rawatkhushu27@gmail.com (K.K.); sanatsujat.singh@csir.res.in (S.S.); sudeshciab1@gmail.com (S.K.Y.); 2Academy of Scientific and Innovative Research (AcSIR), Ghaziabad 201002, India; vishal.acharya@csir.res.in; 3School of Computer Science, UPES University, Dehradun 248007, Uttarakhand, India; samaksh.107995@stu.upes.ac.in (S.M.); vkadyan@ddn.upes.ac.in (V.K.); 4Artificial Intelligence for Computational Biology Lab, Biotechnology Division, CSIR—Institute of Himalayan Bioresource Technology (CSIR-IHBT), Palampur 176061, Himachal Pradesh, India; sharmakritu617@gmail.com; 5Department of Pharmaceutical Sciences, School of Health Sciences and Technology, UPES University, Dehradun 248007, Uttarakhand, India; jupadhyay@ddn.upes.ac.in

**Keywords:** plant tissue culture, plant growth regulators, somatic embryogenesis, artificial intelligence, convolutional neural network

## Abstract

*Ferula assa-foetida* L. (commonly known as asafoetida) is valued for both its medicinal properties and economic significance. However, large-scale cultivation of this plant is challenging due to the prolonged dormancy of its seeds. Recent advancements in computational methods, such as machine learning (ML) and deep learning (DL), have opened new avenues in plant tissue culture research. Studies indicate that these approaches can effectively enhance somatic embryogenesis and shoot organogenesis. In this work, seven ML algorithms and five DL models were compared to identify the most suitable combinations of plant growth regulators (PGRs) for these developmental processes. Among them, the convolutional neural network (CNN) achieved the highest predictive accuracy at 87%. Given that traditional cultivation of *F. assa-foetida* is hindered by seed dormancy, these findings present a promising strategy for establishing reliable in vitro protocols. Such protocols may not only improve propagation through indirect somatic embryogenesis but also contribute to the conservation of this endangered medicinal species.

## 1. Introduction

*Ferula assa-foetida* (L.), commonly known as ‘heeng’ in India, is a well-known medicinal plant with numerous health benefits. It contains several important compounds, such as terpenoids, sulfide derivatives, volatile oils, phenols, and minerals, which confer strong therapeutic potential. Of the 180–185 species in the genus *Ferula*, only *F. assa-foetida* L. produces heeng [1]. Traditionally, asafoetida has served multiple medical applications, including the treatment of respiratory infections, urinary problems, digestive issues, and menstrual regulation. The spice also has dual purposes in medical practice, as it is used to treat snake and insect bites, as well as worm infections [2]. The oleo-gum resin is extracted from the fleshy taproot after five years of cultivation and comprises approximately 25% gum, 40–60% oleo-resin, and 10–17% essential oil [3]. This plant is native to Iran and Afghanistan but was later introduced to other regions. Although it is not originally from India, it has been valued there for centuries, both as a culinary spice and as a component of traditional medicine. In the wild, the plant primarily reproduces via seeds; however, seed dormancy often complicates its propagation [4]. Asafoetida has been classified as an endangered species in Iran’s Red Data Book [4]. Despite its significance, only a limited number of studies have focused on tissue culture and in vitro propagation of *F. assa-foetida* L. (e.g., Zare et al. [5]; Roozbeh et al. [6]). Because conventional propagation is hindered by prolonged seed dormancy [5] and the species is monocarpic (flowering only once before dying), which limits natural multiplication [6]. Therefore, advanced biotechnological approaches are crucial for large-scale propagation and genetic improvement. Among these, in vitro regeneration and genetic transformation hold particular promise for enhancing its cultivation [7]. Plant tissue culture techniques have been widely employed in horticulture, medicine, and the food industry [8]. These methods facilitate the large-scale production of plants with desirable characteristics and provide opportunities for introducing genetic modifications that may enhance their medicinal properties.

The growth of plant cells and organs is significantly influenced by various unpredictable and uncontrollable factors, making outcomes difficult to anticipate. To address this challenge, machine learning (ML) models can be employed to evaluate and optimize culture conditions. As a field within artificial intelligence (AI), ML is emerging as a powerful approach for analysing and improving complex biological systems [9,10]. It assists researchers in making sense of complicated and uncertain data, such as that generated in tissue culture experiments [11]. Recently, advanced ML algorithms, particularly nonlinear ones, have drawn attention in areas such as plant systems biology [12], plant breeding [13], and plant tissue culture [11]. These methods reduce guesswork by identifying patterns in data and suggesting optimal combinations of factors to achieve improved results [14]. Although a relatively new field, ML modelling has limited examples of effective advancements in optimizing plant tissue culture processes, including sterilization, callus formation, in vitro regeneration, and secondary metabolite production [15,16,17].

In light of the above, this research focuses on predicting and optimizing the ideal concentrations and types of plant growth regulators (thidiazuron (TDZ), 6-benzylaminopurine (BAP), and α-naphthaleneacetic acid (NAA)) for successful direct organogenesis by integrating ML and DL models with image-based analysis. It employs seven ML approaches: random forest (RF), support vector machine (SVM), k-nearest neighbours (kNN), decision tree (DT), extreme gradient boosting (XG Boost), naïve bayes, and logistic regression, alongside five DL models: convolutional neural network (CNN), MobileNet, region-based convolutional neural network (RCNN), residual neural network (ResNet), and visual geometry group (VGG19). We propose the application of ML/DL-enabled modelling to accurately predict the optimal PGR levels, thereby enhancing the vigour and replicability of in vitro regeneration in *F. assa-foetida* L.

## 2. Materials and Methods

### 2.1. Callus Induction Under In Vitro Culture Conditions

The research was conducted at the Plant Tissue Culture Laboratory located at CSIR-IHBT in Palampur, Himachal Pradesh. The callus-inducing in vivo leaf explants were obtained from healthy *F. assa-foetida* L. plants (accession no. EC968470) grown in a greenhouse at CSIR-IHBT)). Fully expanded young leaves (2 years old) were collected from the mid-region of the plant during the month of November, when the plants were in their active vegetative growth phase.

The initial step of explant preparation involved surface sterilization to establish aseptic conditions prior to commencing experiments for callus induction. The leaf explants were initially rinsed thoroughly with distilled water to remove dirt and were cleaned using a sable hairbrush with Tween 20 (Himedia, Laboratories Pvt. Ltd., Mumbai, India). Subsequently, the leaves were disinfected with 0.04% (*w*/*v*) streptomycin sulfate and bavistin for 15 min, followed by rinsing with distilled water. Surface sterilisation was then performed in a laminar airflow hood using 0.04% mercuric chloride (*w*/*v*) with a drop of surfactant (diluted Tween 20) for 5 min. Explants were rinsed five to six times with sterilised distilled water. Following this treatment, they were immersed in 70% ethanol (*v*/*v*) for 1 min. Afterward, they were washed again with distilled water and gently dried using Whatman No. 1 filter paper. Explants were cut into approximately 1.0 cm segments and inoculated onto the Murashige and Skoog [18] medium supplemented by a plant growth regulator (PGR).

The MS medium, supplemented with 0.5 mg/L 2,4-dichlorophenoxyacetic acid (2,4-D) and 2.0 mg/L kinetin, successfully induced friable calli cell lines, which were subsequently used for proliferation [19]. The calli cultures underwent regular subculturing every 30–35 days and were maintained in dark conditions at 25 °C with 55% relative humidity.

### 2.2. Media and Culture Conditions

The growth medium consisted of MS medium supplemented with 3% (*w*/*v*) sucrose and 0.78% (*w*/*v*) agar. The pH of the medium was adjusted between 5.70 and 5.75 using 1.0 M NaOH before autoclaving at 121 °C and 15 psi for 15 min. In this experiment, 250 mL Erlenmeyer flasks containing 100 mL of MS medium were used. The experiments were performed in three replicates, each containing three explants, and were repeated three times. MS basal served as the control. Explants used for callus induction were maintained at a temperature of 25 ± 2 °C in the dark, while somatic embryogenesis and shoot induction were observed at the same temperature but under light conditions of 40 μmolm^−2^s^−1^ provided by cool-white LED tubes (Bajaj, Electricals Ltd., Mumbai, India) on a 16/8 h photoperiod.

### 2.3. Somatic Embryogenesis and Shoot Organogenesis from Callus Tissue

Only callus proliferation was observed on the medium containing 0.5 mg/L 2,4-D and 2.0 mg/L kinetin, with no shoot induction occurring even after prolonged subculturing. Therefore, the proliferated calli (0.5 g) was transferred to MS media supplemented with varying concentrations of TDZ (0.2, 0.5, 1.0 and 2.0 mg/L) alone, as well as MS medium containing different BAP concentrations (1.0 mg/L and 2.0 mg/L) along with NAA (0.2 mg/L and 0.5 mg/L), to identify the optimal medium for somatic embryogenesis and shoot initiation and to enhance the number of in vitro-raised shoots. Culture responses to somatic embryos and shoot induction were assessed after 1 month. The morphology of calli tissues and the stages of somatic embryos were examined visually and under a stereomicroscope (Magnus MagCam DC-10, Magnus Opto Systems India Pvt. Ltd., Noida, India).

### 2.4. Dataset Details and Exploratory Data Analysis (EDA)

The dataset for this study comprised high-resolution images and numerical data documenting in vitro shoot organogenesis of *F. assa-foetida* L. under different PGR treatments. Images of shoot development were acquired using a stereozoom microscope (Magnus MagCam DC-10). Numerical data were recorded for each treatment concentration of TDZ, BAP, and NAA. The image dataset comprised 800 RGB images (100 images for each of the 8 PGR concentration combinations), resulting in an exactly balanced 8-class classification problem for PGR concentration prediction.

Experimental treatments were arranged in a Completely Randomised Design (CRD) with 100 replicates per PGR treatment. Data analysis was performed using ORIGIN 2024b, and means were compared using Tukey’s HSD test at α = 0.05. Following data collection, a systematic exploratory data analysis (EDA) was conducted to assess class balance, distribution of shoot counts, and image quality; summary statistics (mean and median shoot counts) for each labelled treatment are reported in Table 1.

### 2.5. Dataset Preprocessing

To ensure accurate model training and reliable predictions, both images and numerical data underwent a thorough preprocessing process. Images were collected from the experiment folder and linked to their respective PGR treatments using a mapping system based on filenames. Each filename contained important information, such as the treatment type and replicate number, which was extracted programmatically. Meanwhile, shoot counts were obtained from organized Excel sheets and matched with the images to create labelled datasets for supervised learning. All images were resized to 32 × 32 pixels (RGB) to balance computational efficiency with the retention of morphological features. To verify that this choice does not harm performance, we performed a resolution comparison at 32 × 32, 64 × 64, and 128 × 128 pixels using the same CNN architecture and training protocol. As summarized in the Appendix A. The 32 × 32 resolution actually yielded the highest mean accuracy (87%) with the lowest training time and number of parameters, whereas 64 × 64 and 128 × 128 resulted in slightly lower accuracies (79% and 78%, respectively) and substantially higher computational cost. Pixel values were normalised by dividing by 255 such that:
$$X_{norm}=\frac{X}{255}$$

where X is the original pixel value matrix, and X_norm_ is scaled to [0, 1]. An image augmentation pipeline including random horizontal and vertical flips and rotations was also evaluated. The same CNN architecture was trained with and without augmentation, and classification accuracy and training time were recorded for both settings (Appendix A). The data was partitioned in an 80:20 train/test ratio, and a validation dataset was held back from the training dataset to be utilized for hyperparameter selection and early stopping during CNN training. Treatments were assigned numeric codes (e.g., TDZ 0.5 mg/L = label 1) for classification tasks. Preprocessed input matrices and target labels were employed to train CNNs and evaluate a range of ML algorithms.

### 2.6. Proposed System Architecture

Initially, high-resolution images of *F. assa-foetida* L. shoot organogenesis were acquired and stored in a curated database, henceforth referred to as the ‘Hinge Image Dataset’ (Figure 1). The dataset was divided into training and test sets using an 80:20 train/test ratio, with a sub-portion of the training set reserved for validation to facilitate hyperparameter optimization and early stopping.

The training set was employed for learning model parameters and recognising visual patterns associated with input PGR treatments, while the validation set guided model selection and helped prevent overfitting. The final holdout test set was used to measure objective performance. A wide variety of ML and deep learning (DL) algorithms were learned and evaluated, including RF, SVM, DT, Logistic Regression, k-NN, XGBoost, Naïve Bayes, CNN, R-CNN, ResNet50, MobileNet V2, and VGG19. Each model was trained independently on the training set (validated on validation), and their performance was compared on the test set. This approach allowed for an exploration of classical ML strategies in comparison to DL architectures for the automated classification of PGR treatments in images and for predicting optimal PGR levels for micropropagation. A CNN was utilised as a DL baseline model; its structure and training regime were designed to extract hierarchical informative image features for discriminating between PGR treatments and to serve as a comparative baseline for evaluating the predictive performance of the other ML/DL approaches. All transfer-learning models (ResNet50, VGG19, MobileNetV2, and the R-CNN variant) were initialized with ImageNet weights (include_top = False) and extended with a GlobalAveragePooling2D layer, a 128-unit ReLU dense layer, and a single output unit. The convolutional backbone was kept frozen, and only the newly added layers were trained using the Adam optimizer and MSE loss with the same early-stopping protocol as the CNN. Light fine-tuning trials were conducted by unfreezing deeper layers; however, due to the small dataset and strong domain mismatch with ImageNet images, these attempts led to rapid overfitting without improving validation accuracy.

### 2.7. Structure of the CNN Model

The CNN employed was hierarchical, with each layer playing a designated role in classification and feature extraction (Figure 2a,b). RGB images of dimensions 32 × 32 × 3, after preprocessing (resizing, changing colour mode, and normalising to the [0, 1] range), were fed into the network. Three levels of convolution using 3 × 3 kernels were employed in the feature extraction process; each convolution layer was accompanied by a Rectified Linear Unit (ReLU) activation function to introduce nonlinearity and enable the network to learn complex relationships. Following each convolution round, a 2 × 2 max-pooling layer reduced spatial dimensions while retaining informative features, thereby decreasing computational load and countering overfitting. The output feature maps were flattened and fed into a fully connected block comprising a dense layer of 128 units to map the extracted features to the output representation. For training purposes, the model was optimised using the Adam optimiser to take advantage of adaptive learning rates and achieve rapid convergence. Mean squared error (MSE) was employed as the loss function, with labels for shoot counts transformed into discrete PGR class labels for classification. Early stopping and validation monitoring were implemented during training to prevent overfitting and to facilitate the selection of the best-performing model based on validation results. The detailed architecture of the CNN is illustrated in Appendix A.

### 2.8. Implementation and Training Strategy

The data was divided into training (80%) and testing (20%) sets using the train-test split method. Early stopping was employed to avoid overfitting by monitoring validation loss and halting training if no improvement was observed. The CNN model was trained for 110 epochs with a batch size of 32 to ensure stable and effective learning. In addition to a single 80:20 train–test split, 5-fold cross-validation were performed using the same CNN architecture and hyperparameters (Appendix A). Model performance was further evaluated using a confusion matrix to identify misclassified PGR concentration classes and to understand error patterns. The dataset was randomly partitioned into five folds with stratification by PGR concentration, and in each run, four folds were used for training, and one fold was used for testing. The mean and standard deviation of classification accuracy across the five folds were computed. All DL models were trained to regress the PGR concentration label (coded as integers 1–8) using mean squared error (MSE) loss. During training, both MSE and the coefficient of determination (R^2^) were monitored to assess convergence and model stability. For evaluation as an 8-class classification task, the continuous outputs of each model were rounded to the nearest integer and compared with the true labels to compute accuracy, precision, recall, F1-score, and to generate the confusion matrix. The CNN and all transfer-learning models were trained exclusively to predict PGR concentration classes (8-class classification) and not shoot counts.

An overview of the complete experimental–computational workflow, from explant preparation and PGR treatments through image acquisition, dataset construction, model training and evaluation, and biological interpretation, is provided in Figure 3.

### 2.9. Computational Environment

All deep learning models were implemented in Python 3.13.7 using TensorFlow/Keras 2.20.0 and were trained on a workstation running Windows 11 equipped with an Intel64 Family 6 Model 186 Stepping 3, GenuineIntel processor (AMD64 architecture, Intel Corporation, Santa Clara, CA, USA) and no dedicated GPU. Approximate training times, input resolutions, and numbers of trainable parameters for the CNN across different image resolutions are summarized in Appendix A, while the corresponding details for the augmentation experiment are presented in Appendix A. The overall hardware and software environment is summarized in Appendix A.

## 3. Results

### 3.1. Effect of TDZ on the Development of Somatic Embryogenesis and Shoot Organogenesis 

The MS medium containing 0.5 mg/L 2,4-D and 2.0 mg/L kinetin effectively promoted the formation of friable callus from leaf explant Figure 4A(a–c)). Changes were observed in calli following the addition of different TDZ concentrations to MS medium. Somatic embryos were noted on calli cultured for 8 weeks on MS media supplemented with varying concentrations of TDZ. On the surface of the calli, globular somatic embryos were observed (Figure 4B(b)). These globular somatic embryos progressed to heart-shaped and ultimately torpedo-stage somatic embryos (Figure 4B(c–g)) after an extended culture period. The embryos exhibited remarkable growth during the first three weeks of culture, adopting a heart-shaped appearance. Subsequently, the somatic embryos continued to develop and entered the torpedo stage over the next four weeks of culture. Shoots were induced in response to various concentrations of TDZ and BAP in combination with NAA, with the highest number of shoots obtained at a TDZ (0.5 mg/L) (Figure 5B(c)). Therefore, this stage was selected for further analysis. These shoots were transferred after one month to half-strength MS medium for root induction, as described by Patial et al. [19]. No shoot organogenesis was observed on MS basal medium. Cytokinins exert a favourable impact on shoot organogenesis and are essential for enhancing shoot multiplication.

### 3.2. Statistical Analysis

A completely randomised design (CRD) with 100 replicates per treatment was employed for the ANOVA. Significant differences between means were determined using Tukey’s HSD test (α = 0.05). TDZ at 0.5 mg/L resulted in significantly higher mean shoot numbers compared to other treatments. Lower and higher concentrations of TDZ produced fewer shoots, suggesting the existence of an optimal concentration range for effective micropropagation. Overall, these results indicate that the response to TDZ at 0.5 mg/L is consistent and reliable for shoot organogenesis. The data obtained from these experiments were subsequently used to create predictive models based on neural networks that could predict effective concentrations of plant growth regulators (PGRs) for in vitro shoot organogenesis.

### 3.3. Performance of Machine Learning Models

The same normalised images were processed using seven classical ML classifiers, reshaped into 1-D feature vectors; reported values represent means on the held-out test set. The highest levels of ML accuracy (82%), along with improved precision and recall, were observed with Logistic Regression and Decision Tree classifiers (Table 2). Random Forest followed closely with 81% accuracy (F1 = 80%), while XGBoost achieved 78% accuracy. Support Vector Machine (SVM) and k-Nearest Neighbours (k-NN) performed poorly (62% and 55% accuracy, respectively), as anticipated, given the difficulty of operating in a high-dimensional flattened image space without extensive feature engineering. Naive Bayes demonstrated moderate competence with 72% accuracy, indicating that a simple probabilistic approach can be effective when the flattened feature distributions have not been neutralised. In summary, the custom CNN proved most effective by leveraging preserved spatial features; however, the reasonable performance of some ML methods suggests that additional feature engineering, dimensionality reduction, or domain-aware preprocessing would benefit non-DL approaches.

### 3.4. Performance of Deep Learning Models

The DL models were tested on the original image tensors, ensuring that spatial structure was maintained during training and inference. A custom CNN, along with three transfer learning networks (MobileNet, ResNet, and VGG19) and an R-CNN, was also tested as an architecture. The highest and most consistent performance was achieved with the custom CNN, which attained 87% accuracy, with equal precision, recall, and F1 values (87%) (Table 2). Across 5-fold cross-validation, the CNN achieved a mean classification accuracy of 80.36 ± 5.19% (Appendix A), which is consistent with the performance observed in the original 80:20 split. This indicates that the model’s performance is robust to the choice of train–test partition. The large differences in shoot counts among treatments (26.21 vs. 0) were not used as ML prediction targets; these values are purely biological measurements. At an input resolution of 32 × 32 pixels, the CNN required approximately 9.37 s of training time and contained 159,944 trainable parameters, achieving an accuracy of 87%. Increasing the resolution to 64 × 64 pixels led to a training time of about 39.26 s with 684,232 parameters and an accuracy of 79%, while 128 × 128 pixels required approximately 140.67 s with 3,305,672 parameters and achieved 78% accuracy (Appendix A). These results justify the choice of 32 × 32 pixels for the final CNN as a good compromise between accuracy and computational cost for the available dataset. In the augmentation experiment, the baseline CNN without augmentation achieved an accuracy of 87%, whereas the augmented model dropped to 13% accuracy despite a training time of 7.82 s (Appendix A). Visual inspection suggested that some augmented images appeared unrealistic relative to the experimental imaging setup, which likely confused the model rather than regularizing it. Consequently, the final CNN model was trained without augmentation. The accuracy of VGG19 was moderate at 75% (F1 = 0.77). In contrast, the transfer learning models (MobileNet, ResNet, and R-CNN) yielded significantly lower accuracy (approximately 26–36%). The consistently lower accuracy of all transfer-learning models (26–36%) appears to stem from the severe domain mismatch between ImageNet natural images and in vitro shoot images. Although we attempted light fine-tuning by unfreezing deeper convolutional layers, these trials led to faster overfitting without improving validation metrics, likely due to the limited dataset size. Consequently, the tailored CNN achieved superior generalization performance. We attribute these poor performance results to the focused, sparse dataset and the intensive fine-tuning of pretrained layers: frozen base layers and feature mismatches between the source-domain images and micropropagation images likely led to underfitting or stalling during testing. These observations suggest that, in specific biological imaging tasks, off-the-shelf transfer learning often requires domain adaptation and substantial task-specific training data to achieve performance comparable to a custom CNN trained directly on the target images. For the CNN model, the regression of true versus predicted PGR concentration on the test set (Figure 6) yielded a mean squared error (MSE) of 0.1697, a root mean squared error (RMSE) of approximately 0.41, and a coefficient of determination (R^2^) of 0.9690, with corresponding training-set values of MSE = 0.0022, RMSE ≈ 0.047, and R^2^ = 0.9996. These regression metrics are consistent with the classification performance obtained after rounding the predictions to the nearest concentration class.

### 3.5. Performance Metrics

The confusion matrix is a crucial tool for evaluating the model’s ability to predict plant growth regulator concentrations from input images. The model identifies visual features of shoot organogenesis and classifies images into different levels of PGR concentration.

In this classification framework:Positive Prediction → The model correctly predicts the actual PGR concentration (Table 3).Negative Prediction → The model incorrectly predicts a different concentration than the actual one.

To evaluate performance, the following metrics were calculated:Accuracy = (TP + TN)/(TP + TN + FP + FN)Precision = TP/(TP + FP)Recall (Sensitivity) = TP/(TP + FN)F1-Score = 2 × (Precision × Recall)/(Precision + Recall)

Accuracy provides an assessment of the percentage of correct predictions. Improved accuracy reveals robust PGR concentration detection with minimal faults.

Precision is defined as true positive predictions divided by actual total predictions, such that higher values reduce false positives.

Recall is defined here as recall of true PGR concentrations when they are present, while minimizing false negatives.

F1-Score balances precision and recall, providing a single comprehensive measure of performance.

The confusion matrix results of the model demonstrated its capability to accurately forecast PGR concentrations from image data. A high TP rate ensured that true concentrations were correctly identified, while a low FN ensured that the model did not frequently overlook true concentrations. The balance between recall and precision resulted in very few misclassifications, underscoring the robustness of the model. Overall, the model exhibited high reliability and accuracy in classifying PGR concentrations from images. This highlights the potential of DL to automatically predict PGR concentrations with minimal human intervention, thereby enhancing the precision of in vitro plant regeneration experiments.

### 3.6. Validation of Performance Metrics

The validation of the experimental results indicated that the CNN model effectively identified the optimal PGRs and their concentrations for shoot induction. Furthermore, the CNN model proved to be a reliable tool for predicting outcomes in in vitro experiments. The best performance was recorded with TDZ at a concentration of 0.5 mg/L, serving as the optimal fitness function for shoot induction (Figure 7b). For the 7th PGR concentration, the CNN correctly classified 23 out of 100 images (23% accuracy) (Figure 7c). The confusion matrix for the CNN classifier (Figure 7c) provides a detailed breakdown of the model’s performance across all eight PGR concentration classes. The rows represent the true PGR concentrations (classes 1–8), while the columns represent the predicted concentrations. The diagonal elements show correctly classified images for each class, whereas off-diagonal elements indicate misclassifications. Classes 1–8 correspond to the following PGR treatments: Class 1: [TDZ (0.5 mg/L)], Class 2: [TDZ (1.0 mg/L)], Class 3: [BAP (2.0 mg/L) + NAA (0.2 mg/L)], Class 4: [BAP (1.0 mg/L) + NAA (0.2 mg/L)], Class 5: [BAP (1.5 mg/L) + NAA (0.5 mg/L)], Class 6: [BAP (2.0 mg/L) + NAA (0.5 mg/L)], Class 7: [TDZ (0.2 mg/L)], Class 8: [TDZ (2.0 mg/L)]. Analysis of the confusion matrix reveals that most misclassifications occur between adjacent PGR concentration classes, particularly among intermediate treatments (classes 3–6), which is consistent with the biological observation that these treatments produce visually similar callus morphologies. In contrast, the extreme treatments (classes 1 and 8, representing the lowest and highest PGR concentrations) show higher classification accuracy, as they exhibit more distinct morphological features. From the confusion matrix, we computed the overall classification accuracy, as well as macro-averaged precision, recall, and F1-score, which are reported in the results. The macro-averaged metrics provide a balanced assessment of performance across all classes, accounting for the equal representation of each PGR concentration in the dataset.

Error analysis using the confusion matrix (Figure 7c) showed that most misclassifications occurred between neighbouring PGR concentrations, particularly among intermediate treatments where callus morphology overlaps. In contrast, extreme treatments (no PGR and the highest PGR level) were classified more accurately. These errors mainly arise when visual differences between treatments are subtle, reflecting the biological variability across intermediate PGR doses.

## 4. Discussion

Despite being an important and valuable spice in Indian cuisine, research interest in the micropropagation of *F. assa-foetida* L. is limited. Over the past decade, there have been few reports on the micropropagation of *Ferula* species. Our study addresses this gap by integrating an experimentally validated in vitro regeneration protocol with AI-based predictive modelling to identify PGR regimes that maximise shoot organogenesis. This combined experimental–computational approach not only confirms previous biological insights but also advances tissue-culture optimization in a manner that is directly actionable for conservation and propagation. Plant regeneration via somatic embryogenesis and organogenesis represents a crucial method for establishing an efficient route to genetic manipulation, thereby fostering advancements at the cellular level [20]. The essential factors ensuring successful plant regeneration include the choice of explant, the composition of the growth medium, and the application of PGRs [21]. To date, there have only been a few published articles on in vitro micropropagation for *Ferula* spp. [6,22,23,24]. Zare et al. [5] reported that hypocotyl-derived callus cultured on MS medium containing 1.0 mg/L BA and 0.2 mg/L NAA produced an average of 7.4 shoots per callus. In contrast, the current study identified TDZ (0.5 mg/L) as the most effective PGR for shoot induction, yielding a significantly higher regeneration response of 26.21 ± 0.42 shoots per callus, demonstrating the superior morphogenic potential of TDZ over BA-NAA combinations in *Ferula assa-foetida*. Roozbeh et al. [6] obtained maximum indirect somatic embryogenesis from root explants cultured on 2,4-D (0.5 mg/L) + kinetin (0.2 mg/L), and the highest embryo-to-seedling conversion occurred on the same PGR combination in B5 medium. Their results contrast with current findings because they used a different explant type and a distinct medium composition. Hasani et al. [25] induced callus from hypocotyl explants using kinetin (0.5–4.0 mg/L) with NAA (0.1–1.0 mg/L), and embryogenic calli formed only after transferring the cultures to hormone-free MS medium. This outcome differs from current results, where TDZ promoted somatic embryogenesis and shoot organogenesis. These differences likely arise from variation in explant type. Together, these studies highlight that regeneration responses in *Ferula assa-foetida* are highly dependent on explant source, basal medium, and PGR composition, which explains the differences observed between previous reports and the current optimized TDZ-based protocol.

The present study utilised varying concentrations of BAP, NAA, and TDZ to effectively promote somatic embryogenesis and shoot organogenesis. Altering the concentrations of BAP, NAA, and TDZ significantly impacted regeneration efficiency and the number of shoots induced. Previous research indicates that TDZ, a phenyl urea derivative, is among the most effective PGRs for in vitro propagation across diverse plant species [26,27,28]. Wu et al. [29] reported that TDZ alone enhanced induction, differentiation, and adventitious shoot proliferation. Generally, TDZ applied at lower concentrations improves plant micropropagation techniques [26]. The effectiveness of shoot organogenesis relies on several factors, including the explant source, medium composition, growth conditions, temperature, etc. In *Ferula jaeschkeana* Vatke, combinations of auxins and cytokinins have been shown to induce callus formation [23]. In a previous study, it was reported that MS medium comprising Kinetin and NAA induces callus initiation, while embryogenic calli were observed in hormone-free medium in *F. assa-foetida* L. [25].

The phenylurea-type cytokinin TDZ functions as a potent cytokinin that activates strong cytokinin signaling pathways through ARABIDOPSIS RESPONSE REGULATORS (ARR-A/ARR-B) to promote cell division, meristem reactivation, and shoot bud formation [30]. The higher doses of TDZ lead to increased oxidative stress and disrupt the natural balance between auxin and cytokinin, which results in vitrification and decreased regeneration success [31]. The 0.5 mg/L concentration of TDZ produces optimal ROS levels while enabling efficient auxin transport and regeneration-related molecular pathway activation, including WUSCHEL and isopentenyltransferase, without disrupting morphogenesis. Research shows TDZ concentrations affect plant development because moderate levels stimulate shoot formation through cytokinin signaling pathways [30] but excessive amounts cause hyperhydricity, induce oxidative stress, and disrupt hormonal equilibrium [32,33,34]. Therefore, the observed optimal performance at 0.5 mg/L aligns well with current understanding of TDZ-mediated cytokinin signaling and stress physiology. Furthermore, BAP enhances cytokinin-mediated cell division through the activation of type-B ARRs, which regulate genes responsible for meristem activity and shoot primordia formation [35]. NAA, a synthetic auxin, facilitates cellular dedifferentiation, maintains callus proliferation, and regulates auxin-responsive factors (ARFs) essential for coordinated morphogenesis [36]. Previous studies have demonstrated that achieving the correct cytokinin-to-auxin ratio is critical, as balanced levels promote organogenic competence. Excess auxin causes excessive callusing and inhibits shoot differentiation, whereas high cytokinin disrupts endogenous auxin gradients, suppresses rooting, and may reduce overall regeneration potential [37,38].

Artificial intelligence (AI) techniques are widely employed across various fields globally; however, their acceptance in plant biotechnology and agriculture has been relatively limited. Over the past few years, artificial neural networks (ANNs) have been utilized in in vitro propagation studies to enhance the accuracy of optimization and result predictions [15,17]. This work is, to our knowledge, one of the first to (i) integrate high-precision image data from explants with shoot counts and (ii) conduct a direct, side-by-side comparison of seven classical ML methods and five DL architectures specifically for predicting regeneration outcomes in plant tissue culture. Convolutional neural networks (CNNs) outperformed classical ML classifiers, achieving 87% classification accuracy compared to approximately 82% for the best ML models (Logistic Regression and Decision Tree). The superior performance of CNNs reflects their capacity to extract complex, hierarchical morphological features from images (such as shape, texture, and subtle growth cues) that simpler feature-based algorithms cannot capture as effectively. We also evaluated transfer learning models (MobileNet, ResNet, R-CNN), which underperformed relative to our CNN trained on the domain-specific dataset, likely due to two reasons: (i) the pre-trained networks were trained on large natural-image corpora whose features do not directly translate to micropropagation imagery (domain mismatch), and (ii) our dataset size limited effective fine-tuning. Together, these findings underscore that bespoke DL training or domain-adapted transfer learning strategies are preferable when working with specialised biological images.

### 4.1. Limitations

Several constraints temper the generalizability of our conclusions. First, feature-visualization methods such as Grad-CAM or saliency maps were not applied; future studies should use these tools to verify that the CNN focuses on biologically meaningful structures, such as callus color and compactness. Second, while 5-fold cross-validation indicates robust performance on the available data, independent external validation on additional biological batches is still required. Third, we did not perform a formal comparison between the model predictions and expert human assessments, which would be an important step towards using such models as decision-support tools in practice. Fourth, all experiments were conducted under controlled in vitro conditions; ex vitro acclimatization and field establishment were not assessed, so downstream survival and performance remain to be validated.

### 4.2. Future Directions

To translate and extend these results, we have three priority paths:Scale and diversify the dataset. Multi-site collaborations and systematic augmentation (controlled image variations, temporal series) will produce larger, more heterogeneous datasets that improve DL generalizability and enable rigorous benchmarking of domain-adapted transfer learning approaches.Broaden experimental scope and mechanistic integration. Expand the PGR search space (additional cytokinins/auxins and finer concentration gradients), and integrate multi-omics (transcriptomics, metabolomics) with imaging data to relate model predictions to underlying biological pathways controlling somatic embryogenesis.Validate and operationalize the pipeline. Test optimized TDZ regimes (including the identified 0.5 mg/L) through acclimatization and field trials, develop interpretable model outputs (saliency maps, feature attribution), and package the imaging + prediction workflow into a reproducible tool for use by tissue-culture laboratories and conservation programs.

By demonstrating that image-driven DL models can accurately predict effective PGR regimes and by pinpointing TDZ 0.5 mg/L as a robust inducer of shoots in *F. assa-foetida*, this study provides a practical, data-driven route for accelerating micropropagation in a species of conservation and medicinal importance. With dataset expansion, mechanistic integration, and ex vitro validation, the framework presented here can be generalized to other recalcitrant medicinal plants and inform large-scale propagation and genetic improvement efforts.

## 5. Conclusions

This study demonstrates that AI-driven modelling significantly enhances the efficiency and optimization of *F. assa-foetida* L. in vitro regeneration. Empirically, TDZ (0.5 mg/L) induced the most prominent shoot induction among the tested treatments. Computationally, CNNs achieved the highest predictive accuracy of 87%, while standard classifiers such as Logistic Regression and Decision Tree realised approximately 82%, showcasing the advantages of DL in extracting complex, nonlinear morphological information from images of explants. The combined image-to-prediction pipeline reduces the need for extensive trial-and-error, thereby lowering material and labour costs and increasing the reproducibility of protocol optimization. This approach, which employs precise phenotyping alongside model-based selection, can be readily adapted for other medicinal or endangered plants that are challenging to propagate. To enhance applicability and stability, future studies should (i) scale and diversify the dataset. Multi-site collaborations and systematic augmentation (e.g., controlled image variations and temporal series) will yield larger, more heterogeneous datasets that improve DL generalisability and enable rigorous benchmarking of domain-adapted transfer learning approaches; (ii) Broaden experimental scope and mechanistic integration. Expanding the PGR search space (including additional cytokinin/auxin and finer concentration gradients) and integrating multi-omics with imaging data will help relate model predictions to the underlying biological pathways controlling somatic embryogenesis. This study illustrates that DL models based on image data can accurately predict the most effective PGR treatments and highlights TDZ (0.5 mg/L) as a strong promoter of shoot organogenesis in *F. assa-foetida* L. With expanded datasets and mechanistic integration, this framework can be extended to other recalcitrant medicinal plants, supporting large-scale propagation and genetic improvement efforts.

## Figures and Tables

**Figure 1 biology-14-01703-f001:**
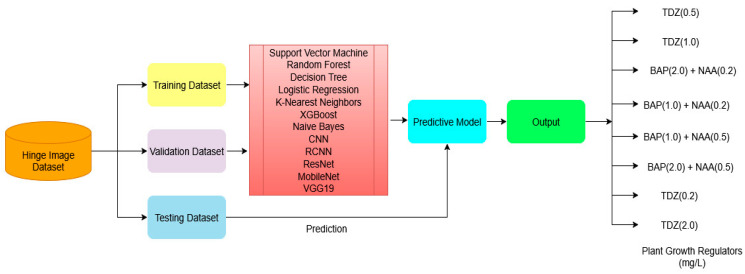
The graphic illustration of the proposed system architecture.

**Figure 2 biology-14-01703-f002:**
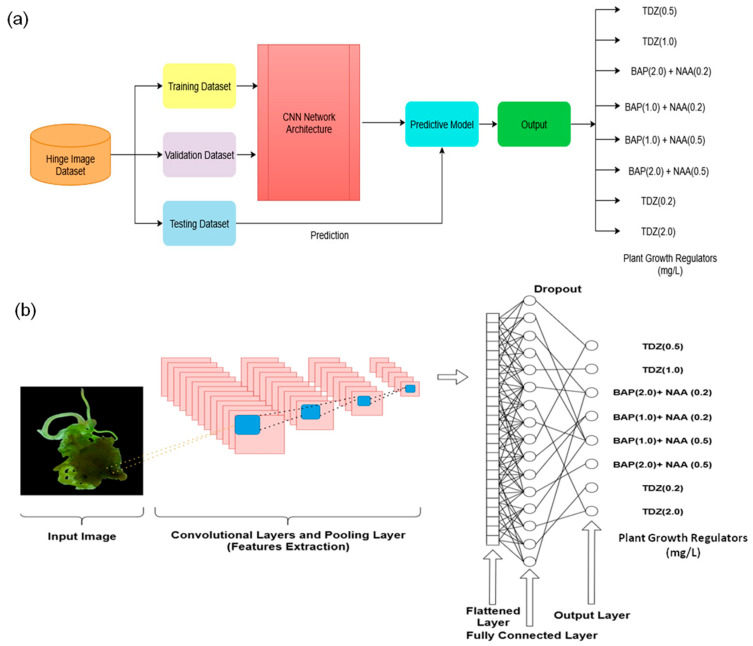
(**a**) CNN model architecture: figure demonstrating the layer-by-layer architecture of the CNN model, with the convolutional operations, pooling functions, and dense layers that make predictions for PGR concentrations; (**b**) Training workflow and feature extraction process: figure emphasizes stepwise methodology followed in training, validation, and classification, where images undergo numerous transformations before they are finally predicted.

**Figure 3 biology-14-01703-f003:**
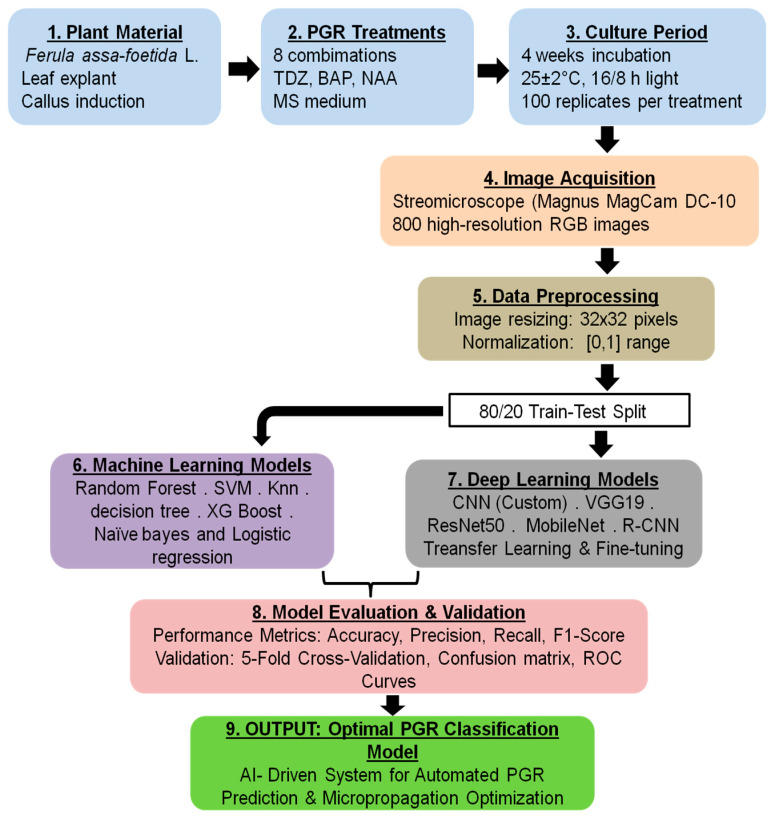
Schematic representation of the experimental–computational pipeline used in the current study. The workflow begins with explant preparation and application of the eight PGR (plant growth regulators) concentration treatments, followed by in vitro culture and standardized image acquisition. The images are organized into a labelled dataset according to PGR concentration, which is used to train and evaluate the CNN and transfer-learning models. The model outputs are then interpreted in the context of shoot regeneration responses to support optimization of PGR treatments.

**Figure 4 biology-14-01703-f004:**
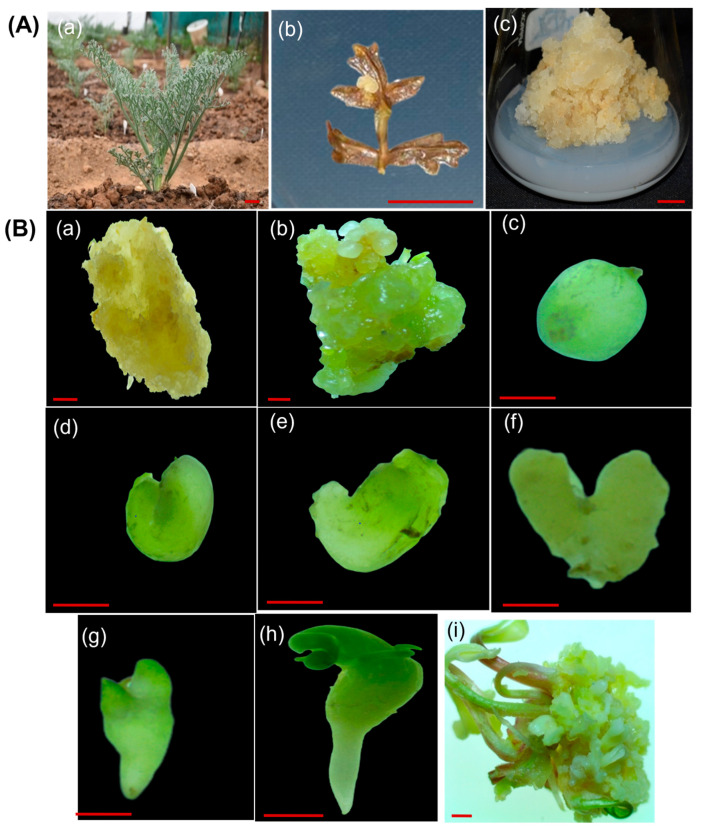
Indirect somatic embryogenesis in *Ferula assa-foetida* L. (**A**) Callus induction and proliferation: (**a**) mother plant, (**b**) callus induction from leaf explant, (**c**) callus proliferation on MS medium supplemented with 0.5 mg/L 2,4-dichlorophenoxyacetic acid (2,4-D) and 2.0 mg/L kinetin. Scale bars: 1 cm (**a**–**c**). (**B**) Stereozoom image of (**a**) Non-embryogenic calli, (**b**) globular calli clump somatic embryo formation from embryogenic calli. (**c**) Globular-shaped somatic embryo; (**d**–**f**) heart-shaped somatic embryo; (**g**) torpedo-shaped somatic embryo; (**h**) cotyledonary somatic embryo. (**i**) Shoot regeneration from embryogenic calli. Scale bars: 1 mm.

**Figure 5 biology-14-01703-f005:**
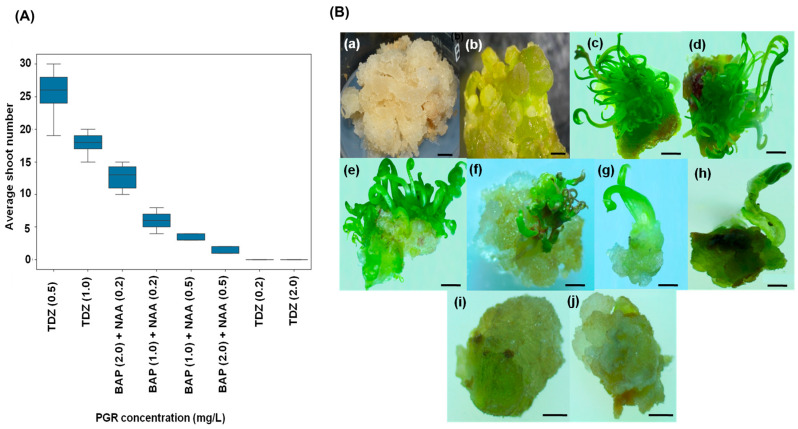
(**A**) Effect of plant growth regulator combinations on multiple shoot induction in *Ferula assa-foetida* L. after one month of culture. Data are mean ± SE (*n* = 100). (**A**) box plot; (**B**) Indirect shoot organogenesis in *Ferula assa-foetida* L.—(**a**) Proliferated friable calli (**b**) Embryogenic calli (scale bar = 2 mm). Shoot induction in different PGRs type—(**c**) TDZ (0.5 mg/L), (**d**) TDZ (1.0 mg/L), (**e**) BAP (2.0 mg/L) + NAA (0.2 mg/L), (**f**) BAP (1.0 mg/L) + NAA (0.2 mg/L), (**g**) BAP (1.0 mg/L) + NAA (0.5 mg/L), (**h**) BAP (2.0 mg/L) + NAA (0.5 mg/L), (**i**) TDZ (0.2 mg/L), (**j**) TDZ (2.0 mg/L). Scale bar: 1 cm (**a**,**i**,**j**); Scale bar: 1 mm (**c**–**h**).

**Figure 6 biology-14-01703-f006:**
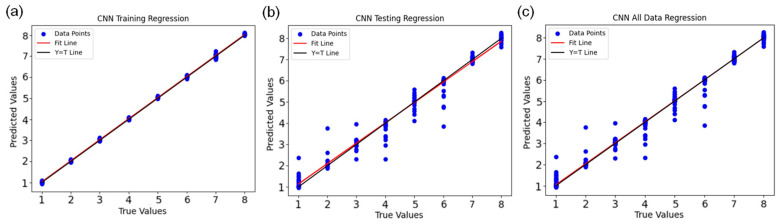
The scatter plot of measured and predicted values of best PGRs in fitted convolutional neural network (CNN) model (**a**) The line plot of measured versus predicted values of PGRs in training stage of CNN (**b**) The line plot of measured versus predicted values of PGRs in testing stage of CNN (**c**) The line plot of all measured versus all predicted values of PGRs by CNN model.

**Figure 7 biology-14-01703-f007:**
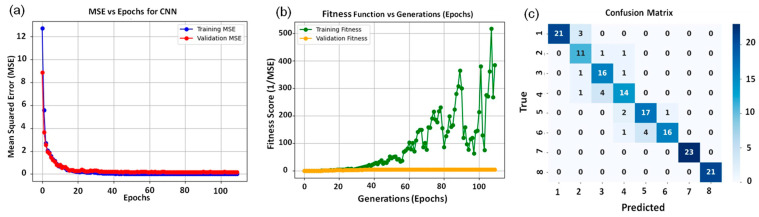
(**a**) The CNN performance based on MSE vs. epochs of PGRs types in *Ferula assa-foetida* L., (**b**) Fitness function score of PGRs value vs. number of optimization generations in *Ferula assa-foetida* L., (**c**) Confusion matrix for CNN classification showing predicted versus true classes. Class labels correspond to PGR concentrations as follows: 1—TDZ (0.5 mg/L); 2—TDZ (1.0 mg/L); 3—BAP (2.0 mg/L) + NAA (0.2 mg/L); 4—BAP (1.0 mg/L) + NAA (0.2 mg/L); 5—BAP (1.5 mg/L) + NAA (0.5 mg/L); 6—BAP (2.0 mg/L) + NAA (0.5 mg/L); 7—TDZ (0.2 mg/L); 8—TDZ (2.0 mg/L).

**Table 1 biology-14-01703-t001:** Eight different PGR concentrations were used in the model and graphs.

Concentration (mg/L)	Mean Shoot Count	Median of Shoot Count	Number of Images
TDZ (0.5)	26.21	26	100
TDZ (1.0)	17.88	18	100
BAP (2.0) + NAA (0.2)	12.73	13	100
BAP (1.0) + NAA (0.2)	6.2	6	100
BAP (1.0) + NAA (0.5)	3.49	3	100
BAP (2.0) + NAA (0.5)	1.47	1	100
TDZ (0.2)	0	0	100
TDZ (2.0)	0	0	100

**Table 2 biology-14-01703-t002:** Comparative performance of seven ML models and five DL models on shoot induction (accuracy, precision, recall, F1-score).

Model	Accuracy (%)	Precision (%)	Recall (%)	F1-Score (%)
Random Forest	81	83	81	80
SVM	62	66	62	62
k-Nearest Neighbours	55	60	54	52
Decision Tree	82	80	80	80
XGBoost	78	80	77	77
Naïve Bayes	72	72	71	70
Logistic Regression	82	82	82	81
CNN	87	87	87	87
MobileNet	26	41	26	26
R-CNN	35	46	35	35

**Table 3 biology-14-01703-t003:** Performance Metrics Classification Framework.

	Correctly Predicted (Positive)	Incorrectly Predicted (Negative)
True PGR Concentration	True Positive (TP): Model correctly predicts the actual concentration.	False Negative (FN): Model fails to predict the true concentration.
False PGR Concentration	False Positive (FP): Model incorrectly assigns a wrong concentration.	True Negative (TN): Model correctly identifies absence of a concentration.

## Data Availability

Data will be shared when required.

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
