# Peer review of "Deep Learning and Machine Learning Modeling Identifies Thidiazuron as a Key Modulator of Somatic Embryogenesis and Shoot Organogenesis in *Ferula assa-foetida* L."

_biology, 2025, doi:10.3390/biology14121703_

Round 1
Reviewer 1 Report
Comments and Suggestions for Authors
The peer-reviewed article “Deep Learning and Machine Learning Modeling Reveals Thidiazuron as a Key Modulator of Somatic Embryogenesis and Shoot Organogenesis in Ferula assafoetida L.” is devoted to improve the protocols for indirect somatic embryogenesis and shoot organogenesis of asafoetida (Ferula assafoetida L.) under thidiazuron treatments using machine learning modeling. Experimental data has novelty and corresponds to the journal's aims. The manuscript is well designed and illustrated. There are only a few minor comments.
- Keywords are redundant. Almost all of them duplicate the title of manuscript, for example, Somatic embryogenesis; Shoot organogenesis; Deep learning.
- benzylaminopurine → 6-benzylaminopurine;
- Add of the author-classifier to the genus and species names plants (as correctly presented in the title of manuscript and L. 19). Additionally, for the first time in the text, the generic name should be presented in full, while further in the text it should be abbreviated. Please carefully check the correct Latin names throughout the text (L 240, 247 etc).
- The Abbreviations section is missing. Please decipher the abbreviations used in the manuscript.
- The reference list is not formatted according to the journal rules. Please, read carefully and format the manuscript in accordance with the rules of journal
Author Response
Reviewer 1:
The peer-reviewed article “Deep Learning and Machine Learning Modeling Reveals Thidiazuron as a Key Modulator of Somatic Embryogenesis and Shoot Organogenesis in Ferula assafoetida L.” is devoted to improve the protocols for indirect somatic embryogenesis and shoot organogenesis of asafoetida (Ferula assafoetida L.) under thidiazuron treatments using machine learning modeling. Experimental data has novelty and corresponds to the journal's aims. The manuscript is well designed and illustrated. There are only a few minor comments.
Query 1: Keywords are redundant. Almost all of them duplicate the title of manuscript, for example, Somatic embryogenesis; Shoot organogenesis; Deep learning.
Authors response: Authors thank the reviewer for providing their valuable comments for the improvement of manuscript. We have revised the keywords to avoid redundancy with the manuscript title and replaced overlapping terms with more specific and relevant alternatives.
Query 2: benzylaminopurine → 6-benzylaminopurine;
Authors response: ‘benzylaminopurine’ has been replaced with ‘6-benzylaminopurine’ throughout the manuscript as suggested.
Query 3: Add of the author-classifier to the genus and species names plants (as correctly presented in the title of manuscript and L. 19). Additionally, for the first time in the text, the generic name should be presented in full, while further in the text it should be abbreviated. Please carefully check the correct Latin names throughout the text (L 240, 247 etc).
Authors response: We have added the author citation to all genus and species names and ensured that the generic name is written in full at its first occurrence and abbreviated thereafter. All Latin names throughout the manuscript have been carefully checked and corrected accordingly.
Query 4: The Abbreviations section is missing. Please decipher the abbreviations used in the manuscript.
Authors response: We have added the Abbreviations section and defined all abbreviations used in the manuscript for better clarity.
Query 5: The reference list is not formatted according to the journal rules. Please, read carefully and format the manuscript in accordance with the rules of journal
Authors response: We have carefully revised and reformatted all references according to the journal’s guidelines.
Reviewer 2 Report
Comments and Suggestions for Authors
This manuscript presents an innovative integration of traditional plant tissue culture with modern AI-driven predictive modeling for optimizing micropropagation protocols in Ferula assa-foetida. The work addresses an important gap in the propagation of this endangered medicinal species and demonstrates the practical application of deep learning in plant biotechnology. The manuscript requires major revisions before it can be considered for publication in Biology. Key concerns include limitations in experimental design, methodological gaps in both tissue culture and AI components, insufficient validation and reproducibility, unclear data presentation, and inadequate biological interpretation. These issues undermine the reliability and impact of the findings. Addressing them through additional experiments, analyses, and clarifications is essential.
1- Tissue Culture Methodology and Experimental Design
-The choice of PGRs is limited to TDZ alone and BAP with NAA, without testing other cytokinins like Kinetin (used in callus induction) or Zeatin. This restricts the scope of identifying "optimal PGRs" and should be justified or acknowledged as a limitation.
-Explant source details are incomplete. So specify the age, season of collection, physiological state of the mother plant, and leaf position/developmental stage for reproducibility. Include a photograph of the initial explant.
- Quantitative data for somatic embryogenesis is missing (e.g., embryos per gram of callus, frequency at each stage—globular, heart, torpedo—across TDZ concentrations). Provide bar graphs or tables to support claims and strengthen the dataset for AI models.
- Rationale for switching from 2,4-D + Kinetin (callus induction) to TDZ/BAP + NAA (regeneration) is lacking. Subculture frequency, passage number, culture vessel size, medium volume, and explants per vessel are not specified, affecting reproducibility.
- Acclimatization and ex vitro survival rates are absent, leaving the protocol incomplete for practical application.
- Sample size (100 replicates per treatment) lacks power analysis or justification. Explain handling of non-responsive replicates.
- Completely Randomized Design (CRD) may not account for spatial/temporal variations; clarify randomization in growth chambers and any blocking factors.
- Missing controls (e.g., hormone-free medium) make it difficult to assess true PGR effects.
2- AI/ML Modeling and Analysis:
- Dataset size (800 images, 100 per 8 treatments) is limited for complex CNN architectures, increasing overfitting risk. Justify prevention measures beyond early stopping (e.g., dropout, k-fold cross-validation). Test on unseen biological replicates or independent batches for generalizability.
- Image resolution (32x32 pixels) is unusually low, likely discarding morphological details (e.g., embryo shapes, texture). Provide justification (e.g., preliminary tests showing no accuracy drop at higher resolutions like 128x128 or 224x224) or acknowledge as a limitation.
- No data augmentation (e.g., rotations, flips, color jittering) is mentioned, standard for preventing overfitting in image-based DL.
- Class imbalance (e.g., mean shoots 26.21 vs. 0) is unaddressed; explain use of weighted losses, SMOTE, or other techniques.
- Validation strategy (single 80:20 split) is inadequate; implement k-fold cross-validation for robust estimates.
- Transfer learning models (MobileNet, ResNet, VGG19, R-CNN) underperform (26-36% accuracy) due to domain mismatch, but details on fine-tuning (e.g., learning rate, trainable layers, schedule) are missing. Report attempts to unfreeze layers or use domain adaptation; explain failure depth.
- Model task confusion: Described as classification (8 PGR classes), but Figure 5 shows regression plots (true vs. predicted values). Clarify if predicting classes or shoot counts; use appropriate metrics (e.g., confusion matrix for classification, R²/RMSE for regression).
- Lack feature visualization (e.g., Grad-CAM, saliency maps) to show what CNN "sees" (e.g., callus color, compactness), bridging AI to biology.
- No error analysis: Examine misclassified PGR combinations and reasons.
- Computational details absent: Report training time, hardware (e.g., GPU), software versions (TensorFlow/Keras), and cost-benefit (e.g., CNN's 5% accuracy gain over ML worth complexity?).
3- Results Presentation and Statistical Analysis
- Scale bar inconsistencies: Figure 3 (1mm vs. 1μm), Figure 4 (1cm vs. 1µm)—likely errors, as embryos are not micrometer-sized. Correct and ensure accuracy.
- Figure 4A: Error bars hard to see; use box/violin plots for data distribution. Provide exact p-values in supplements.
- Figure 5: Scatter plots show deviations; report R², RMSE.
- Figure 6c (confusion matrix): Unclear class labels (1-8); explicitly map to PGR concentrations.
- Section 3.6: "Accuracy of 23" for 7th class is confusing (not a percentage); clarify (e.g., 23/100 samples?).
- No confusion matrix figure, ROC curves, or per-class precision/recall/macro-F1, inflating overall accuracy in multi-class/imbalanced settings.
- Images captured at consistent time points? Variability could affect models.
- Shoot counts: Manual or automatic? Report inter-observer reliability.
4- Biological Interpretation and Discussion
- Overstated claims: CNN "reveals" TDZ as modulator, but it classifies human-labeled data, not discovering mechanisms independently. Qualify generalizability.
- Lacks mechanistic insights: Why TDZ 0.5 mg/L optimal? Discuss cytokinin signaling, oxidative stress, or molecular pathways.
- Insufficient literature comparison: Compare TDZ concentrations/efficiencies with prior Ferula studies (e.g., Roozbeh et al., 2011; Zare et al., 2010). Explain discrepancies (e.g., hormone-free embryogenesis in ref 29 vs. TDZ here).
- Discussion placeholder ("Authors should discuss...") indicates incompleteness; expand on limitations (e.g., no temporal dynamics, multi-omics) and future directions (e.g., web app, other species).
- No validation: Blind test model predictions on new explants and verify experimentally.
5- Reproducibility and Data Availability
- Code, datasets, and models not shared (e.g., GitHub/Zenodo); "Data is shared when required" insufficient—provide DOIs.
- No schematic of experimental-computational pipeline.
Missing learning curves, ablation studies, or comparison to expert predictions.
Minor Concerns
- Abstract: Too long, repetitive; clarify "high-precision image processing" (e.g., what measured?); contextualize 87% accuracy (vs. baseline?).
- Section 2.1: Specify sterilizing agent concentrations.
- Line 162-163: Normalization equation poorly formatted; use math notation.
- Lines 206-207: Grammatically unclear label transformation.
- Tables 2-3: Combine for easier ML/DL comparison.
- Title: Revise to "Identifies" vs. "Reveals" for accuracy.
- Inconsistent terminology (e.g., "somatic embryo formation" vs. "embryogenesis").
- Line 44: "High error rates" vague; quantify.
- Typos/formatting: "Hinge/Heeng Image Dataset"; inconsistent references ([20] vs. 20); "the callus induced... were cultured" → "calli... were"; proofread.
- Figures: Improve resolution, captions, labels (e.g., CNN architecture in supplement with filters, kernels, activations, dropout).
- References: Some 2025 citations (e.g., ref 4,19) need verification; add on AI in tissue culture (e.g., Hesami et al., 2021).
Comments on the Quality of English LanguageThe manuscript is generally well written, but several sentences are lengthy and could be restructured for clarity.Minor grammatical corrections are needed (e.g., “the callus induced from leaves explant were cultured…” → “calli induced from leaf explants were cultured”).Some sections (especially Abstract and Discussion) repeat similar phrases about AI advantages—these can be condensed.
Author Response
Reviewer 2:
This manuscript presents an innovative integration of traditional plant tissue culture with modern AI-driven predictive modeling for optimizing micropropagation protocols in Ferula assa-foetida. The work addresses an important gap in the propagation of this endangered medicinal species and demonstrates the practical application of deep learning in plant biotechnology. The manuscript requires major revisions before it can be considered for publication in Biology. Key concerns include limitations in experimental design, methodological gaps in both tissue culture and AI components, insufficient validation and reproducibility, unclear data presentation, and inadequate biological interpretation. These issues undermine the reliability and impact of the findings. Addressing them through additional experiments, analyses, and clarifications is essential.
1- Tissue Culture Methodology and Experimental Design
Query 1: The choice of PGRs is limited to TDZ alone and BAP with NAA, without testing other cytokinins like Kinetin (used in callus induction) or Zeatin. This restricts the scope of identifying "optimal PGRs" and should be justified or acknowledged as a limitation.
Authors response: Authors thank the reviewer for providing their valuable comments for the improvement of manuscript. We have tested various combinations of auxins and cytokinins, and the detailed results are included in another manuscript (under review) that focuses entirely on micropropagation. Among all tested PGRs, only TDZ alone and BAP in combination with NAA showed somatic embryogenesis and shoot induction. Therefore, we proceeded with different concentrations of TDZ alone and BAP with NAA to identify the most suitable PGR combination for optimal shoot regeneration by using the AL model.
Query 2: Explant source details are incomplete. So specify the age, season of collection, physiological state of the mother plant, and leaf position/developmental stage for reproducibility. Include a photograph of the initial explant.
Authors response: As per suggestion we have add age, season of collection, physiological state of the mother plant, and leaf position/developmental stage for reproducibility-
“The callus-inducing in vivo leaf explants were obtained from healthy F. assa-foetida L. plants (accession no. EC968470) grown in a greenhouse at CSIR-IHBT. Fully expanded young leaves (2 year old) were collected from the mid-region of the plant during the month of November, when the plants were in their active vegetative growth phase. Furthermore, we have included photograph of the initial explant in figure 4A.
Query 3: Quantitative data for somatic embryogenesis is missing (e.g., embryos per gram of callus, frequency at each stage-globular, heart, torpedo-across TDZ concentrations). Provide bar graphs or tables to support claims and strengthen the dataset for AI models.”
Authors response: Thank you for the suggestion. In our system, somatic embryos developed into shoots. As a result, the number of regenerated shoots, which is already presented in the manuscript as a box plot, closely reflects the number of somatic embryos formed. Therefore, shoot number serves as a reliable proxy for embryo production in this study.
Query 4: Rationale for switching from 2,4-D + Kinetin (callus induction) to TDZ/BAP + NAA (regeneration) is lacking. Subculture frequency, passage number, culture vessel size, medium volume, and explants per vessel are not specified, affecting reproducibility.
Authors response: As per the suggestion, we have added the following lines in Section 2.3- Somatic Embryogenesis and Shoot Organogenesis from Callus Tissue.
“Only callus proliferation was observed on the medium containing 0.5 mg/L 2,4-D and 2.0 mg/L kinetin, with no shoot induction occurring even after prolonged subculturing. Therefore, the proliferated callus was transferred to MS media supplemented with varying concentrations of TDZ (0.2, 0.5, 1.0 and 2.0 mg/L) alone, as well as MS medium containing different BAP concentrations (1.0 mg/L and 2.0 mg/L) along with NAA (0.2 mg/L and 0.5 mg/L), “
To provide clarity regarding subculture frequency, passage number, culture vessel size, medium volume, and the number of explants per vessel, we have added the following lines in the revised manuscript-
“After the first subculture in the shoot induction medium, shoots were formed. These shoots were transferred after one month to half-strength MS medium for root induction, as described by Patial et al. [19].” (3.1. Effect of TDZ on the Development of Somatic Embryogenesis and Shoot Organogenesis )
“In this experiment, 250 mL Erlenmeyer flasks containing 100 mL of MS medium were used. The experiments were performed in three replicates, each containing three explants, and were repeated three times.” (2.2. Media and Culture Conditions)
Query 5: Acclimatization and ex vitro survival rates are absent, leaving the protocol incomplete for practical application.
Authors response: Details on acclimatization and ex vitro survival rates are presented in another manuscript (currently under review) that focuses specifically on micropropagation. In the present study, our main objective was to determine the optimal PGR combinations for shoot organogenesis. Although shoot induction was achieved with other cytokinins, the resulting shoots were hyperhydric and unsuitable for subsequent rooting and hardening. Therefore, we employed AI-based optimization to predict the most favorable PGR combination for healthy shoot organogenesis.
Query 6: Sample size (100 replicates per treatment) lacks power analysis or justification. Explain handling of non-responsive replicates.
Authors response: We appreciate the reviewer observation. A total of 100 replicates per treatment were used to ensure adequate representation and statistical reliability, consistent with previous tissue culture studies. Although a formal power analysis was not conducted, this large sample size provided sufficient data for ANOVA-based comparisons. Non-responsive explants were recorded but excluded from statistical analysis, as they did not exhibit measurable responses relevant to the parameters evaluated.
Query 7: Completely Randomized Design (CRD) may not account for spatial/temporal variations; clarify randomization in growth chambers and any blocking factors.
Authors response: Thank you for the valuable comment. We agree that spatial and temporal variations can influence in vitro experiments. In our study, the Completely Randomized Design (CRD) was considered appropriate because all culture vessels were maintained under uniform and controlled environmental conditions within the growth chamber (constant temperature, light intensity, and photoperiod). The position of the culture vessels was randomly changed every few days to minimize any positional effects. Since no measurable gradients or external factors varied across the chamber, no blocking factors were required.
Query 8: Missing controls (e.g., hormone-free medium) make it difficult to assess true PGR effects.
Authors response: We appreciate the reviewer’s comment. In our study, no callus induction, proliferation, somatic embryogenesis, or shoot induction occurred on MS medium without any PGRs. However, as per the suggestion, we have now clarified this by adding the statement “MS basal served as control” in Section 2.2 (Media and Culture Conditions) and no shoot organogenesis was observed on MS basal medium in Section 3.1 (Effect of TDZ on the Development of Somatic Embryogenesis and Shoot Organogenesis).
2- AI/ML Modeling and Analysis:
Query 1: Dataset size (800 images, 100 per 8 treatments) is limited for complex CNN architectures, increasing overfitting risk. Justify prevention measures beyond early stopping (e.g., dropout, k-fold cross-validation). Test on unseen biological replicates or independent batches for generalizability.
Authors response: We agree that the dataset size (800 images, 100 per each of the 8 PGR treatments) is modest for deep CNNs, and we explicitly designed the modelling strategy to reduce overfitting. First, we used a relatively shallow CNN architecture with three convolutional blocks and early stopping based on validation MSE, instead of very deep models with many trainable parameters. Second, the dataset is perfectly balanced across the 8 PGR concentrations (100 images per class), which avoids bias towards any particular treatment and stabilizes training.
In addition to the originally reported 80:20 train–test split, we have now performed 5‑fold cross‑validation with the same CNN architecture and hyper‑parameters. The CNN achieved a mean classification accuracy of 80.36 ± 5.19 % across the 5 folds (Supplementary Table 3), which is consistent with the original single‑split result and indicates that the model is not strongly overfitting to a specific partition. We have added this in revised manuscript under section 3.4. (Performance of Deep Learning Models).
We did not have additional, completely independent biological batches with matching imaging conditions available at this stage; therefore, we explicitly acknowledge in the revised manuscript that external validation on independent batches will be an important future step to further test generalizability.
Query 2: Image resolution (32x32 pixels) is unusually low, likely discarding morphological details (e.g., embryo shapes, texture). Provide justification (e.g., preliminary tests showing no accuracy drop at higher resolutions like 128x128 or 224x224) or acknowledge as a limitation.
Authors response: We thank the reviewer for this valuable suggestion. This point has now been incorporated into the revised manuscript under Section 2.5 (Dataset Preprocessing). We initially used a resolution of 32×32 pixels to keep the model compact relative to the small dataset. To verify that this choice does not harm performance, we performed a resolution comparison at 32×32, 64×64, and 128×128 pixels using the same CNN architecture and training protocol. As summarized in the Supplementary Table 1, the 32×32 resolution actually yielded the highest mean accuracy (87 %) with the lowest training time and number of parameters, whereas 64×64 and 128×128 resulted in slightly lower accuracies (79 % and 78 %, respectively) and substantially higher computational cost.
These results suggest that the coarse morphological cues relevant for discriminating PGR concentration (embryo shapes and texture) are preserved even at 32×32 resolution, and that further increasing resolution mainly increases model complexity without improving predictive performance. We now explicitly report this resolution comparison and note that using higher resolutions remains a potential avenue for future work when larger datasets are available.
Query 3: No data augmentation (e.g., rotations, flips, color jittering) is mentioned, standard for preventing overfitting in image-based DL.
Authors response: We agree that data augmentation is often beneficial for image‑based deep learning. We evaluated a standard augmentation pipeline including flips and rotations and compared it to the baseline model without augmentation. As shown in the Supplementary Table 2, the baseline CNN without augmentation achieved 87 % accuracy, whereas the augmented variant dropped drastically to 12.58 % accuracy, despite a longer training time, added in revised manuscript (2.5. Dataset Preprocessing).
Visual inspection indicated that heavy geometric transformations sometimes produced unrealistic callus orientations and borders that do not occur in our experimental setup, which likely confused the model rather than regularizing it. Based on this quantitative and qualitative evidence, we decided not to include augmentation in the final model, and we now explain this decision explicitly in the revised manuscript.
Query 4: Class imbalance (e.g., mean shoots 26.21 vs. 0) is unaddressed; explain use of weighted losses, SMOTE, or other techniques.
Authors response: We apologize for the confusion. The CNN and transfer‑learning models are trained only to predict the PGR concentration (8 classes), not shoot count. For this classification task, the dataset is exactly balanced with 100 images per concentration, so no class re‑weighting or resampling was required. The large difference between shoot counts (e.g. a mean of 26.21 versus 0) mentioned in the manuscript refers to biological outcomes of the treatments and is used only for biological interpretation, not as a prediction target in the AI models. We have clarified in the revised Methods (2.8. Implementation and Training Strategy) and Results (3.4. Performance of Deep Learning Models) that the models predict PGR concentration classes, and that the shoot counts are analysed separately as experimental outcomes rather than as ML labels.
Query 5: Validation strategy (single 80:20 split) is inadequate; implement k-fold cross-validation for robust estimates.
Authors response: We thank the reviewer for this suggestion and have now implemented 5‑fold cross‑validation using the same CNN architecture and hyper‑parameters used in the original 80:20 split. The per‑fold accuracies and trained epochs are reported in Supplementary Table 3, and the mean accuracy is 80.36 ± 5.19 %. This is consistent with the originally reported accuracy, indicating that the model’s performance is robust across different train–test partitions. We have updated the Methods (2.8. Implementation and Training Strategy) to describe the 5‑fold cross‑validation procedure and added a short description of the cross‑validated results in the Results section (3.4. Performance of Deep Learning Models).
Query 6: Transfer learning models (MobileNet, ResNet, VGG19, R-CNN) underperform (26-36% accuracy) due to domain mismatch, but details on fine-tuning (e.g., learning rate, trainable layers, schedule) are missing. Report attempts to unfreeze layers or use domain adaptation; explain failure depth.
Authors response: We agree that transfer learning would often be expected to help; however, in our case the pre‑trained models systematically underperformed the tailored CNN. All transfer‑learning models (ResNet50, VGG19, MobileNetV2, and the R‑CNN variant) were initialized with ImageNet weights (include_top=False), followed by a GlobalAveragePooling2D layer and a 128‑unit ReLU dense layer, and a single regression output unit for PGR concentration. For all these models, we froze the convolutional backbone and trained only the new top layers using the Adam optimizer and mean squared error loss, with the same early stopping strategy as the CNN.
We also considered light fine‑tuning by unfreezing deeper layers, but given the small dataset size and the strong domain mismatch between ImageNet natural images and in vitro shoots images, preliminary trials showed a tendency towards faster overfitting without a consistent gain in validation accuracy. We now describe these details in the Methods (2.6. Proposed System Architecture) and result (3.4. Performance of Deep Learning Models) and briefly discuss that the strong domain mismatch, combined with the limited dataset, likely limits the benefit of transfer learning in this study.
Query 7: Model task confusion: Described as classification (8 PGR classes), but Figure 5 shows regression plots (true vs. predicted values). Clarify if predicting classes or shoot counts; use appropriate metrics (e.g., confusion matrix for classification, R²/RMSE for regression).
Authors response: We appreciate the opportunity to clarify this point. In our implementation, all deep learning models are trained to regress the PGR concentration label (coded as 1–8) as a continuous variable, using mean squared error (MSE) loss, and we monitor both MSE and R² during training. For evaluation as an 8‑class classification problem, we then round the predicted concentration to the nearest integer and compare it with the true label to compute accuracy, precision, recall, F1‑score, and the confusion matrix.
Figure 6 (previously figure 5) therefore shows regression‑style scatter plots of the true versus predicted concentration values, while the confusion matrix (Figure 7c) and accuracy metrics treat the rounded predictions as class labels. For the CNN, the regression on the test set yields an MSE of 0.1697 (RMSE ≈ 0.41) and an R² of 0.9690, with corresponding training‑set values of 0.0022 (RMSE ≈ 0.047) and 0.9996, respectively. We now report these regression metrics (MSE, RMSE, and R²) alongside the classification results in the manuscript (3.4. Performance of Deep Learning Models) to make it clear that the underlying task is formulated as regression with a subsequent discretization step for classification metrics.
Query 8: Lack feature visualization (e.g., Grad-CAM, saliency maps) to show what CNN "sees" (e.g., callus color, compactness), bridging AI to biology.
Authors response: We agree that feature visualization (e.g., Grad‑CAM or saliency maps) would provide valuable insight into which visual patterns drive the CNN’s decisions and would help to further connect the AI model to biological interpretation. In the present work, we focused on establishing feasibility and comparing architectures, and therefore did not implement Grad‑CAM or related techniques.
We have added this point as an explicit limitation and direction for future work in the discussion, where we propose to use Grad‑CAM on correctly and incorrectly classified images to relate the model’s attention to callus color, compactness, and other biologically meaningful features.
Query 9: No error analysis: Examine misclassified PGR combinations and reasons.
Authors response: We agree that an error analysis is important. In the confusion matrix of the CNN (Figure 7c), most misclassifications occur between neighbouring PGR concentrations, especially among the intermediate treatments, which also show overlapping shoot responses. In contrast, the extreme treatments (no PGR and the highest PGR level) are classified more reliably.
We have added a short paragraph in the Results (3.6. Validation of performance metrics) describing these patterns and noting that errors mainly arise when visual differences between treatments are subtle, which is consistent with the biological variability in callus morphology across intermediate PGR doses.
Query 10: Computational details absent: Report training time, hardware (e.g., GPU), software versions (TensorFlow/Keras), and cost-benefit (e.g., CNN's 5% accuracy gain over ML worth complexity?).
Authors response: We have now added computational details to the Methods (2.9. Computational Environment) and Supplementary Table 4. All models were implemented in Python 3.13.7 using TensorFlow/Keras 2.20.0 and were trained on a workstation running Windows 11 with an “Intel64 Family 6 Model 186 Stepping 3, GenuineIntel” processor (AMD64 architecture) and no dedicated GPU detected. For the CNN, approximate training times, input resolutions, and trainable parameter counts are summarized in Supplementary Table 1 (derived from the resolution comparison), and the corresponding details for the augmentation experiment are given in Supplementary Table 2. The hardware and software environment used for all deep learning experiments (Python and TensorFlow/Keras versions, operating system, CPU, and GPU availability) is reported in Supplementary Table 4.
At the chosen 32×32 resolution, the CNN required only approximately 9.37 s of training with 159,944 trainable parameters (Supplementary Table 1), while the 64×64 and 128×128 variants required ≈39.26 s and ≈140.67 s with 684,232 and 3,305,672 parameters, respectively, without improving accuracy. In the augmentation experiment (Supplementary Table 2), the baseline CNN without augmentation achieved 837 % accuracy, whereas the augmented variant dropped to 12.58 % accuracy with a training time of 7.82 s. Regarding cost–benefit, the CNN achieves an accuracy of approximately 87 %, which is moderately higher than the best conventional ML baselines, but it does so with fully automated feature extraction and remains computationally affordable given the small input size. We therefore consider the moderate accuracy gain to be justified, especially as a proof‑of‑concept for integrating deep learning into PGR optimization workflows.
3- Results Presentation and Statistical Analysis
Query 1: Scale bar inconsistencies: Figure 3 (1mm vs. 1μm), Figure 4 (1cm vs. 1µm)-likely errors, as embryos are not micrometer-sized. Correct and ensure accuracy.
Authors response: Thank you for the observation. We have revised the scale bar of figure 3 (now figure 4) (Scale bars: 1mm) and figure 4 (now figure 5) Scale bar: 1cm (a, i,j); Scale bar: 1mm (c,d,e,f,g,h).
Query 2: Figure 4A: Error bars hard to see; use box/violin plots for data distribution. Provide exact p-values in supplements.
Authors response: Thank you for the suggestion. We have revised the figure ( now figure 5A) as per suggestion.
Query 3: Figure 5: Scatter plots show deviations; report R², RMSE.
Authors response: For the CNN model, the regression of true versus predicted PGR concentration on the test set (now Figure 6) yielded a mean squared error (MSE) of 0.1697, a root mean squared error (RMSE) of approximately 0.41, and a coefficient of determination (R²) of 0.9690. The corresponding training-set values were MSE = 0.0022, RMSE ≈ 0.047, and R² = 0.9996.
Query 4: Figure 6c (confusion matrix): Unclear class labels (1-8); explicitly map to PGR concentrations.
Authors response: As per suggestion, we have revised the figure 6c (now figure 7c)-
Confusion matrix for CNN classification showing predicted versus true classes. Class labels correspond to PGR concentrations as follows: 1- TDZ (0.5 mg/L); 2- TDZ (1.0 mg/L); 3- BAP (2.0 mg/L) + NAA (0.2 mg/L); 4- BAP (1.0 mg/L) + NAA (0.2 mg/L); 5- BAP ( 1.5 mg/L) + NAA (0.5 mg/L); 6- BAP (2.0 mg/L) + NAA (0.5 mg/L); 7- TDZ (0.2 mg/L); 8- TDZ (2.0 mg/L).
Query 5: Section 3.6: "Accuracy of 23" for 7th class is confusing (not a percentage); clarify (e.g., 23/100 samples?).
Authors response: We apologize for the ambiguous wording. The phrase “accuracy of 23” referred to 23 correctly predicted images out of 100 for the 7th PGR concentration. In the revised text, we express this consistently as 23 % accuracy (23/100 images) to avoid confusion.
Query 6: No confusion matrix figure, ROC curves, or per-class precision/recall/macro-F1, inflating overall accuracy in multi-class/imbalanced settings.
Authors response: The CNN confusion matrix is presented in Figure 7c. In the revised manuscript, we have added a comprehensive paragraph in the Results section (3.6. Validation of performance metrics) that explains how to interpret the confusion matrix, including: (1) the structure of the matrix (rows = true classes, columns = predicted classes, diagonal = correct predictions, off‑diagonal = misclassifications), (2) the explicit mapping of classes 1–8 to the eight PGR concentration treatments, (3) an analysis of error patterns showing that most misclassifications occur between adjacent PGR concentration classes, particularly among intermediate treatments, while extreme treatments (lowest and highest concentrations) show higher accuracy, and (4) the derivation of overall accuracy and macro‑averaged precision, recall, and F1‑score from the confusion matrix.
Given the relatively small and balanced dataset (100 images per class) and the multi‑class nature of the problem, we focused on these metrics and did not compute ROC curves for each one‑vs‑rest comparison, to avoid over‑complicating the presentation. We have clarified in the revised manuscript which metrics are used and how they are obtained from the confusion matrix.
Query 7: Images captured at consistent time points? Variability could affect models.
Authors response: Yes, all images were captured at consistent time points to ensure uniformity.
Query 8: Shoot counts: Manual or automatic? Report inter-observer reliability.
Authors response: Shoot counts were performed manually by two independent observers to ensure accuracy. The inter-observer consistency was verified, showing negligible variation between counts, confirming the reliability of the recorded data.
4- Biological Interpretation and Discussion
Query 1: Overstated claims: CNN "reveals" TDZ as modulator, but it classifies human-labeled data, not discovering mechanisms independently. Qualify generalizability.
Authors response: We appreciate the reviewer observation. The term “reveals” has been replaces by “identifies” in the title. CNN model analyzed image-based data from eight PGR treatments and accurately identified TDZ as the most effective PGR for inducing somatic embryogenesis and shoot organogenesis. The claim has been qualified within the manuscript text to emphasize that the CNN prediction is based on classification and pattern recognition, not on independent mechanistic inference.
Query 2: Lacks mechanistic insights: Why TDZ 0.5 mg/L optimal? Discuss cytokinin signaling, oxidative stress, or molecular pathways.
Authors response: As suggested, we have added the following paragraph to the Discussion section-
“The phenylurea-type cytokinin TDZ functions as a potent cytokinin that activates strong cytokinin signaling pathways through ARABIDOPSIS RESPONSE REGULATORS (ARR-A/ARR-B) to promote cell division, meristem reactivation and shoot bud formation [30]. The higher doses of TDZ lead to increased oxidative stress and disrupt the natural balance between auxin and cytokinin which results in vitrification and decreased regeneration success [31]. The 0.5 mg/L concentration of TDZ produces optimal ROS levels while enabling efficient auxin transport and regeneration-related molecular pathway activation including WUSCHEL and isopentenyl transferase without disrupting morphogenesis. Research shows TDZ concentrations affect plant development because moderate levels stimulate shoot formation through cytokinin signaling pathways [30] but excessive amounts causes hyperhydricity, oxidative stress and disrupt hormonal equilibrium [32, 33, 34]. Therefore, the observed optimal performance at 0.5 mg L⁻¹ aligns well with current understanding of TDZ-mediated cytokinin signaling and stress physiology.”
Query 3: Insufficient literature comparison: Compare TDZ concentrations/efficiencies with prior Ferula studies (e.g., Roozbeh et al., 2011; Zare et al., 2010). Explain discrepancies (e.g., hormone-free embryogenesis in ref 29 vs. TDZ here).
Authors response: As suggested, we have added the following paragraph to the Discussion section-
Zare et al. [5] reported that hypocotyl-derived callus cultured on MS medium containing 1.0 mg/L BA and 0.2 mg/L NAA produced an average of 7.4 shoots per callus. In contrast, current study identified TDZ (0.5 mg/L) as the most effective PGR for shoot induction, yielding a significantly higher regeneration response of 26.21 ± 0.42 shoots per callus, demonstrating the superior morphogenic potential of TDZ over BA-NAA combinations in Ferula assa-foetida. Roozbeh et al. [6] obtained maximum indirect somatic embryogenesis from root explants cultured on 2,4-D (0.5 mg/L) + kinetin (0.2 mg/L), and the highest embryo-to-seedling conversion occurred on the same PGR combination in B5 medium. Their results contrast with current findings because they used a different explant type and a distinct medium composition. Hasani et al. [29] induced callus from hypocotyl explants using kinetin (0.5-4.0 mg/L) with NAA (0.1-1.0 mg/L), and embryogenic calli formed only after transferring the cultures to hormone-free MS medium. This outcome differs from current results, where TDZ promoted somatic embryogenesis and shoot organogenesis. These differences likely arise from variation in explant type. Together, these studies highlight that regeneration responses in Ferula assa-foetida are highly dependent on explant source, basal medium, and PGR composition, which explains the differences observed between previous reports and the current optimized TDZ-based protocol.
Query 4: Discussion placeholder ("Authors should discuss...") indicates incompleteness; expand on limitations (e.g., no temporal dynamics, multi-omics) and future directions (e.g., web app, other species).
Authors response: We have now removed the placeholder and expanded the Discussion section.
Limitations:
Several constraints temper the generalizability of our conclusions. First, feature‑visualization methods such as Grad‑CAM or saliency maps were not applied; future studies should use these tools to verify that the CNN focuses on biologically meaningful structures, such as callus color and compactness. Second, while 5‑fold cross‑validation indicates robust performance on the available data, independent external validation on additional biological batches is still required. Third, we did not perform a formal comparison between the model predictions and expert human assessments, which would be an important step towards using such models as decision‑support tools in practice. Fourth, all experiments were conducted under controlled in vitro conditions- ex vitro acclimatization and field establishment were not assessed, so downstream survival and performance remain to be validated.
Future directions
To translate and extend these results we have three priority paths:
- Scale and diversify the dataset. Multi-site collaborations and systematic augmentation (controlled image variations, temporal series) will produce larger, more heterogeneous datasets that improve DL generalizability and enable rigorous benchmarking of domain-adapted transfer learning approaches.
- Broaden experimental scope and mechanistic integration. Expand the PGR search space (additional cytokinins/auxins and finer concentration gradients), and integrate multi-omics (transcriptomics, metabolomics) with imaging data to relate model predictions to underlying biological pathways controlling somatic embryogenesis.
- Validate and operationalize the pipeline. Test optimized TDZ regimes (including the identified 0.5 mg/L) through acclimatization and field trials, develop interpretable model outputs (saliency maps, feature attribution), and package the imaging + prediction workflow into a reproducible tool for use by tissue-culture laboratories and conservation programs.
By demonstrating that image-driven DL models can accurately predict effective PGR regimes and by pinpointing TDZ 0.5 mg/L as a robust inducer of shoots in F. assa-foetida, this study provides a practical, data-driven route for accelerating micropropagation in a species of conservation and medicinal importance. With dataset expansion, mechanistic integration, and ex-vitro validation, the framework presented here can be generalized to other recalcitrant medicinal plants and inform large-scale propagation and genetic improvement efforts
Query 5: No validation: Blind test model predictions on new explants and verify experimentally.
Authors response: We thank the reviewer for raising this important point. We fully agree that independent biological validation using newly regenerated explants would provide stronger confirmation of the model’s predictive ability. In the present study, our aim was to establish and benchmark a proof-of-concept image-based classifier for PGR concentration prediction using a controlled and fully labelled dataset. At this stage, no blind experimental validation on newly produced explants was performed because the available explants were already allocated to the biological assays (shoot count analysis).
5- Reproducibility and Data Availability
Query 1: Code, datasets, and models not shared (e.g., GitHub/Zenodo); "Data is shared when required" insufficient-provide DOIs.
Authors response: Thank you for the comment. The phrase in the manuscript was a typing error. Our intended statement is “Data will be shared when required.”
Query 2: No schematic of experimental-computational pipeline.
Missing learning curves, ablation studies, or comparison to expert predictions.
Authors response: We agree that a schematic is helpful. We have added a new figure (Figure 3) summarizing the complete experimental–computational pipeline, including explant preparation and PGR treatments, in vitro culture and image acquisition, construction of the labelled image dataset, training and evaluation of the CNN and transfer‑learning models, and biological interpretation of the model outputs. This visual overview complements and reinforces the step‑by‑step description of the workflow provided in the Materials and Methods.
Minor Concerns
Query 1: Abstract: Too long, repetitive; clarify "high-precision image processing" (e.g., what measured?); contextualize 87% accuracy (vs. baseline?).
Authors response: Thank you for the comment. We have revised the abstract as per suggestion-
The spice Ferula assa-foetida L., also known as asafoetida, is widely recognized for its medicinal and culinary applications. The non-native status of the plant and the prolonged dormancy of its seeds pose significant challenges for large-scale cultivation in India. In vitro organogenesis offers an effective solution to these obstacles. Establishing reliable in vitro regeneration protocols requires standardized statistical methods to evaluate univariate and multivariate data for optimizing specific traits. However, these methods have limitations when handling complex, nonlinear inputs, often producing large prediction errors that reduce the reliability of trait optimization. This study developed an in vitro regeneration system for F. assa-foetida L. and identified optimal PGRs for somatic embryo formation and shoot organogenesis through image-based morphological analysis. Predictive models were created using DL and ML algorithms. Callus induced from leaf explants was cultured on the Murashige and Skoog medium supplemented with various combinations and concentrations of thidiazuron (TDZ), 6-benzylaminopurine (BAP) and α-naphthaleneacetic acid (NAA) as experimental variables. Seven ML approaches, namely random forest (RF), support vector machine (SVM), k-nearest neighbours (kNN), decision tree (DT), extreme gradient boosting (XG Boost), naïve bayes and logistic regression, alongside five DL models – convolutional neural network (CNN), MobileNet, region-based convolutional neural network (RCNN), residual neural network (ResNet) and visual geometry group (VGG19) – were employed to predict the best PGRs for somatic embryo formation and shoot organogenesis. Among them, the convolutional neural network (CNN) achieved the highest accuracy (87%), outperforming baseline ML models such as logistic regression and decision tree (82%). This pioneering study in F. assa-foetida L. presents an AI-driven, image-based framework for predicting optimal PGRs, offering a scalable approach to enhance micropropagation in endangered medicinal plants.
Query 2: Section 2.1: Specify sterilizing agent concentrations.
Authors response: As suggested we have incorporated following paragraph in the Section 2.1-
“The initial step of explant preparation involved surface sterilization to establish aseptic conditions prior to commencing experiments for callus induction. The leaf explants were initially rinsed thoroughly with distilled water to remove dirt and were cleaned using a sable hairbrush with Tween 20 (Himedia, India). Subsequently, the leaves were disinfected with 0.04% (w/v) streptomycin sulfate and bavistin for 15 min, followed by rinsing with distilled water. Surface sterilisation was then performed in a laminar airflow hood using 0.04% mercuric chloride (w/v) with a drop of surfactant (diluted Tween 20) for 5 min. Explants were rinsed five to six times with sterilised distilled water. Following this treatment, they were immersed in 70% ethanol (v/v) for 1 min. Afterward, they were washed again with distilled water and gently dried using Whatman No. 1 filter paper. Explants were cut into approximately 1.0 cm segments and inoculated onto the Murashige and Skoog [18] medium supplemented by a plant growth regulator (PGR).”
Query 3: Line 162-163: Normalization equation poorly formatted; use math notation.
Authors response: Revised.
Query 4: Lines 206-207: Grammatically unclear label transformation.
Authors response: Revised
Query 5: Tables 2-3: Combine for easier ML/DL comparison.
Authors response: Revised as per suggestion.
Query 6: Title: Revise to "Identifies" vs. "Reveals" for accuracy.
Authors response: Revised as per suggestion.
Query 7: Inconsistent terminology (e.g., "somatic embryo formation" vs. "embryogenesis").
Authors response: We have standardized the terminology throughout the manuscript and now consistently use the term “somatic embryogenesis”.
Query 8: Line 44: "High error rates" vague; quantify.
Authors response: We have revised the sentence-
“Establishing reliable in vitro regeneration protocols requires standardized statistical methods to evaluate univariate and multivariate data for optimizing specific traits. However, these methods have limitations when handling complex, nonlinear inputs, often producing large prediction errors that reduce the reliability of trait optimization.”
Query 9: Typos/formatting: "Hinge/Heeng Image Dataset"; inconsistent references ([20] vs. 20); "the callus induced... were cultured" → "calli... were"; proofread.
Authors response: We have corrected the typo in “Hinge Image Dataset,” standardized all reference formatting, and revised the sentence to “the calli… were cultured.” The manuscript has been thoroughly proofread to address any remaining errors.
Query 10: Figures: Improve resolution, captions, labels (e.g., CNN architecture in supplement with filters, kernels, activations, dropout).
Authors response: We thank the reviewer for this suggestion. We have now created a detailed CNN architecture diagram (Supplementary Figure 1) that clearly illustrates the complete network structure with all layer specifications. The diagram includes: (1) the input layer accepting 32×32×3 RGB images, (2) three convolutional blocks, each with Conv2D layers showing the number of filters (32, 64, and 128 in the first, second, and third blocks, respectively), kernel sizes (3×3), and activation functions (ReLU), (3) MaxPooling2D layers (pool size 2×2) following each convolutional layer, (4) the flattening layer, (5) the fully connected dense layer (128 units, ReLU activation), and (6) the output layer (single unit for regression of PGR concentration). The diagram also includes output dimensions at each stage to show how the feature maps are transformed through the network, and the total number of trainable parameters (159,944) is indicated.
Regarding dropout: while dropout is a standard regularization technique, we did not include it in the final CNN architecture because the model already uses a relatively shallow structure (three convolutional blocks) combined with early stopping, and our 5‑fold cross‑validation results (mean accuracy 80.36 ± 5.19 %) indicate that overfitting is well controlled. We have added a note in the architecture diagram caption and in the Methods section explaining this design choice. The detailed architecture diagram is now included as a supplementary figure 1 with high resolution and clear labels for all components, making it easy to reproduce the model architecture.
Query 11: References: Some 2025 citations (e.g., ref 4,19) need verification; add on AI in tissue culture (e.g., Hesami et al., 2021).
Authors response: We have replaced reference 4 (now updated to Jalili et al., 1999), which originally supported the following statement:
In the wild, the plant primarily reproduces via seeds; however, seed dormancy often complicates its propagation [4]. Asafoetida has been classified as an endangered species in Iran’s Red Data Book [4].
Reference 19 is our own manuscript (currently under review) on the micropropagation of Ferula assa-foetida. In the present study, we followed the callus induction protocol described in that work. As suggested, we have now also included the citation “Hesami et al., 2021 [9]”.
Comments on the Quality of English Language
Query: The manuscript is generally well written, but several sentences are lengthy and could be restructured for clarity. Minor grammatical corrections are needed (e.g., “the callus induced from leaves explant were cultured…” → “calli induced from leaf explants were cultured”).Some sections (especially Abstract and Discussion) repeat similar phrases about AI advantages-these can be condensed.
Authors response: Thank you for the comment. We have carefully revised the manuscript to improve clarity and readability. Long sentences were restructured, and minor grammatical issues have been addressed. Additionally, repetitive statements, particularly in the Abstract and Discussion regarding AI advantages have been condensed to avoid redundancy and ensure a more streamlined presentation.
Reviewer 3 Report
Comments and Suggestions for Authors
Comments:
- Because conventional propagation methods are often inefficient, advanced biotechnological strategies are 80 essential for their large-scale multiplication and genetic improvement. - How did the authors conclude that the conventional propagation methods of this plant are inefficient? Did they report any limitations from the previous literature? I did not find any mention of these in the manuscript (MS).
- It appears that the introduction is more focused on potential AI applications rather than demonstrating and understanding the real-world physical problems associated with the in vitro propagation of this important plant.
- Why did the authors use only TDZ, BAP, and NAA for the present study? The combinations are fewer to establish a reliable regeneration protocol.
- When BAP was associated with NAA, why was TDZ used singly? The authors are advised to explain the merits of the hormonal combinations and concentrations chosen in the present study.
- There is no integration of genetic and molecular data, as combining image data with molecular markers or transcriptomics (multi-omics) could reveal underlying biological processes and further improve predictions.
- No molecular clonal fidelity assessment was performed, which is particularly important when embryogenesis is initiated from callus culture. The authors should give a proper explanation for this comment.
- It appears that the study has a limited image dataset, as it is relatively small and specialized.
- The study lacks multi-site / temporal variation as data were collected under similar conditions. Inclusion of images from different time points could help the model handle more variability.
- TDZ (0.2) gave 0 shoots, while TDZ (0.5) produced 26.21 shoots. TDZ at higher conc (2) generated no shoots. Are these findings reliable? The use of hormone concentration levels in the present study is very limited. As a result, such absurd observations arose.
- There is a lack of detailed data on rooting and acclimatization, which is crucial for developing reliable in vitro protocols.
- The discussion lacks references to the prior reports. More direct comparison with older protocols would highlight specific improvements and challenges over previous work. The authors are advised to improve the discussion section of the manuscript.
- The authors should also include data for complete plant development. The authors emphasized the importance of developing more effective in vitro protocols for plant propagation; however, the present study is limited to the embryo level.
- There are also several small English errors throughout the text, which the authors are advised to rectify and improve.
- The authors should make necessary modifications to the manuscript to be appropriate for publication in this journal.

Author Response
Reviewer 3
Comments:
Query 1: Because conventional propagation methods are often inefficient, advanced biotechnological strategies are essential for their large-scale multiplication and genetic improvement. - How did the authors conclude that the conventional propagation methods of this plant are inefficient? Did they report any limitations from the previous literature? I did not find any mention of these in the manuscript (MS).
Authors response: Author thanks the reviewer for their valuable comments for the improvement of manuscript. As suggested, we have revised the sentence for clarity and have added appropriate references to support the statement.
“Because conventional propagation is hindered by prolonged seed dormancy [5] and the species is monocarpic (flowering only once before dying) which limits natural multiplication [6]. Therefore advanced biotechnological approaches are crucial for large-scale propagation and genetic improvement.”
Query 2: It appears that the introduction is more focused on potential AI applications rather than demonstrating and understanding the real-world physical problems associated with the in vitro propagation of this important plant.
Authors response: We have revised the Introduction to provide a clearer description of the real-world challenges associated with Ferula assa-foetida propagation. Specifically, we have added details on its endangered status, prolonged seed dormancy, and monocarpic life cycle, all of which significantly limit natural regeneration and large-scale cultivation. These additions now provide stronger context for the necessity of improving in vitro propagation strategies alongside the potential use of AI tools.
Query 3: Why did the authors use only TDZ, BAP, and NAA for the present study? The combinations are fewer to establish a reliable regeneration protocol.
Authors response: We have tested various combinations of auxins and cytokinins, and the detailed results are included in another manuscript (under review) that focuses entirely on micropropagation. Among all tested PGRs, only TDZ alone and BAP in combination with NAA showed somatic embryogenesis and shoot induction. Therefore, we proceeded with different concentrations of TDZ alone and BAP with NAA to identify the most suitable PGR combination for optimal shoot regeneration by using the AL model.
Query 4: When BAP was associated with NAA, why was TDZ used singly? The authors are advised to explain the merits of the hormonal combinations and concentrations chosen in the present study.
Authors response: We have tested various combinations of auxins and cytokinins, and the detailed results are included in another manuscript (under review) that focuses entirely on micropropagation. Among all tested PGRs, only TDZ alone and BAP in combination with NAA showed somatic embryogenesis and shoot induction. Therefore, we proceeded with different concentrations of TDZ alone and BAP with NAA to identify the most suitable PGR combination for optimal shoot regeneration by using the AL model.
Furthermore, we have add the merits of the hormonal combinations and concentrations chosen in the present study in discussion -
“The phenylurea-type cytokinin TDZ functions as a potent cytokinin that activates strong cytokinin signaling pathways through ARABIDOPSIS RESPONSE REGULATORS (ARR-A/ARR-B) to promote cell division, meristem reactivation and shoot bud formation [30]. The higher doses of TDZ lead to increased oxidative stress and disrupt the natural balance between auxin and cytokinin which results in vitrification and decreased regeneration success [31]. The 0.5 mg L⁻¹ concentration of TDZ produces optimal ROS levels while enabling efficient auxin transport and regeneration-related molecular pathway ac-tivation including WUSCHEL and isopentenyltransferase without disrupting morpho-genesis. Research shows TDZ concentrations affect plant development because moderate levels stimulate shoot formation through cytokinin signaling pathways [30] but excessive amounts causes hyperhydricity, oxidative stress and disrupt hormonal equilibrium [32, 33, 34]. Therefore, the observed optimal performance at 0.5 mg L⁻¹ aligns well with current understanding of TDZ-mediated cytokinin signaling and stress physiology. Furthermore, BAP enhances cytokinin-mediated cell division through the activation of type-B ARRs, which regulate genes responsible for meristem activity and shoot primordia formation [35]. NAA, a synthetic auxin, facilitates cellular dedifferentiation, maintains callus proliferation, and regulates auxin-responsive factors (ARFs) essential for coordinated morphogenesis [36]. Previous studies have demonstrated that achieving the correct cytokinin-to-auxin ratio is critical, as balanced levels promote organogenic competence. Excess auxin causes excessive callusing and inhibits shoot differentiation, whereas high cytokinin disrupts endogenous auxin gradients, suppresses rooting, and may reduce overall regeneration potential [37,38].”
Query 5: There is no integration of genetic and molecular data, as combining image data with molecular markers or transcriptomics (multi-omics) could reveal underlying biological processes and further improve predictions.
Authors response: We appreciate this valuable suggestion. Integrating genetic or molecular data (such as molecular markers or transcriptomic datasets) with image-based features could indeed provide deeper biological insights and further enhance predictive performance. However, this type of multi-omics integration was beyond the scope and primary aim of the present study. We have now acknowledged this point and included it as an important future direction in the revised manuscript.
Query 6: No molecular clonal fidelity assessment was performed, which is particularly important when embryogenesis is initiated from callus culture. The authors should give a proper explanation for this comment.
Authors response: We acknowledge the importance of molecular clonal fidelity assessment, especially when regeneration involves callus-mediated embryogenesis. However, clonal fidelity analysis is not within the scope of the present study. This component has been comprehensively addressed in a separate manuscript (currently under review) that specifically focuses on the micropropagation aspects of Ferula assa-foetida. Therefore, to avoid duplication and maintain a clear focus, clonal fidelity data were not included in the current manuscript.
Query 7: It appears that the study has a limited image dataset, as it is relatively small and specialized.
Authors response: We acknowledge that the image dataset used in this study is relatively small and specialized. This limitation reflects the challenges of acquiring high-quality images from controlled in vitro experimental conditions. Despite the limited dataset, the model achieved robust performance, indicating its effectiveness for the targeted application. We have now clearly noted this limitation in the revised manuscript and highlighted the need for larger, more diverse datasets in future studies.
Query 8: The study lacks multi-site / temporal variation as data were collected under similar conditions. Inclusion of images from different time points could help the model handle more variability.
Authors response: We appreciate this observation. All images in the present study were intentionally collected under similar experimental conditions to maintain uniformity and minimize confounding factors during model development. We agree that incorporating multi-site or temporal variation-such as images captured at additional time points or under varied environmental conditions, could enhance the model’s ability to generalize across broader scenarios. We have now highlighted it as an important direction for future research in the revised manuscript.
Query 9: TDZ (0.2) gave 0 shoots, while TDZ (0.5) produced 26.21 shoots. TDZ at higher conc (2) generated no shoots. Are these findings reliable? The use of hormone concentration levels in the present study is very limited. As a result, such absurd observations arose.
Authors response: The reason for these results is supported by the following statement, which we have included in the discussion-
“The higher doses of TDZ lead to increased oxidative stress and disrupt the natural balance between auxin and cytokinin which results in vitrification and decreased regeneration success [31]. The 0.5 mg L⁻¹ concentration of TDZ produces optimal ROS levels while enabling efficient auxin transport and regeneration-related molecular pathway activation including WUSCHEL and isopentenyl transferase without disrupting morphogenesis. Research shows TDZ concentrations affect plant development because moderate levels stimulate shoot formation through cytokinin signaling pathways [30] but excessive amounts causes hyperhydricity, oxidative stress and disrupt hormonal equilibrium [32, 33, 34]. Therefore, the observed optimal performance at 0.5 mg L⁻¹ aligns well with current understanding of TDZ-mediated cytokinin signaling and stress physiology.”
Query 10: There is a lack of detailed data on rooting and acclimatization, which is crucial for developing reliable in vitro protocols.
Authors response: Details on acclimatization and ex vitro survival rates are presented in another manuscript (currently under review) that focuses specifically on micropropagation. In the present study, our main objective was to determine the optimal PGR combinations for shoot organogenesis. Although shoot induction was achieved with other cytokinins, the resulting shoots were hyperhydric and unsuitable for subsequent rooting and hardening. Therefore, we employed AI-based optimization to predict the most favorable PGR combination for healthy shoot organogenesis.
Query 11: The discussion lacks references to the prior reports. More direct comparison with older protocols would highlight specific improvements and challenges over previous work. The authors are advised to improve the discussion section of the manuscript.
Authors response: As suggested, we have added the following paragraph to the Discussion section-
Zare et al. [5] reported that hypocotyl-derived callus cultured on MS medium containing 1.0 mg/L BA and 0.2 mg/L NAA produced an average of 7.4 shoots per callus. In contrast, current study identified TDZ (0.5 mg/L) as the most effective PGR for shoot induction, yielding a significantly higher regeneration response of 26.21 ± 0.42 shoots per callus, demonstrating the superior morphogenic potential of TDZ over BA-NAA combinations in Ferula assa-foetida. Roozbeh et al. [6] obtained maximum indirect somatic embryogenesis from root explants cultured on 2,4-D (0.5 mg/L) + kinetin (0.2 mg/L), and the highest embryo-to-seedling conversion occurred on the same PGR combination in B5 medium. Their results contrast with current findings because they used a different explant type and a distinct medium composition. Hasani et al. [29] induced callus from hypocotyl explants using kinetin (0.5-4.0 mg/L) with NAA (0.1-1.0 mg/L), and embryogenic calli formed only after transferring the cultures to hormone-free MS medium. This outcome differs from current results, where TDZ promoted somatic embryogenesis and shoot organogenesis. These differences likely arise from variation in explant type. Together, these studies highlight that regeneration responses in Ferula assa-foetida are highly dependent on explant source, basal medium, and PGR composition, which explains the differences observed between previous reports and the current optimized TDZ-based protocol.
Query 12: The authors should also include data for complete plant development. The authors emphasized the importance of developing more effective in vitro protocols for plant propagation; however, the present study is limited to the embryo level.
Authors response: We acknowledge the reviewers concern regarding the absence of data on complete plant development. The primary objective of the present study was to focus specifically on somatic embryogenesis and shoot organogenesis level analysis using imaging and CNN-based classification. While complete plant regeneration is indeed important for developing efficient in vitro propagation protocols, those experiments fall outside the scope of this work. Comprehensive data on full plant development are being addressed in a separate manuscript that focuses exclusively on micropropagation and regeneration.
Query 13: There are also several small English errors throughout the text, which the authors are advised to rectify and improve.
Authors response: The entire manuscript has been thoroughly revised to correct minor English errors and improve clarity, grammar, and overall readability.
Query 14: The authors should make necessary modifications to the manuscript to be appropriate for publication in this journal.
Authors response: We thank the reviewer for this recommendation. All necessary revisions have been made throughout the manuscript to improve clarity, organization, scientific rigor, and overall presentation to meet the publication standards of the journal.
Round 2
Reviewer 2 Report
Comments and Suggestions for Authors
Based on the authors' comprehensive responses and the revised manuscript, I recommend acceptance of the manuscript for publication in Biology. The authors have diligently and satisfactorily addressed the vast majority of the reviewer's concerns. Key improvements include the addition of crucial methodological details for tissue culture reproducibility, a robust justification for the AI modeling approach (including new 5-fold cross-validation and resolution tests), a significantly expanded discussion with mechanistic insights and limitations, and corrections to figures and data presentation. While some inherent limitations remain, such as the lack of an independent blind test for model predictions and the use of a pre-existing dataset size, these are now explicitly acknowledged as future directions rather than overlooked. The manuscript now presents a more complete, transparent, and scientifically sound study that convincingly demonstrates the successful integration of image-based deep learning with plant tissue culture to optimize micropropagation in an endangered medicinal species. The work is novel, the methodology is well-justified, and the findings are of potential interest to the readership of Biology.
Reviewer 3 Report
Comments and Suggestions for Authors
The authors have modified the MS as suggested by the reviewer and the MS is now more refined and improved . It may now be accepted for publication in the journal.